# Recent Advances in Light-Driven Semiconductor-Based Micro/Nanomotors: Optimization Strategies and Emerging Applications

**DOI:** 10.3390/molecules29051154

**Published:** 2024-03-05

**Authors:** Vanessa R. A. Ferreira, Manuel A. Azenha

**Affiliations:** CIQUP—Institute of Molecular Sciences, Departamento de Química e Bioquímica, Faculdade de Ciências da Universidade do Porto, Rua do Campo Alegre, 4169-007 Porto, Portugal

**Keywords:** light-driven motors, semiconductor, self-propulsion, photocatalysis, environment, biomedicine

## Abstract

Micro/nanomotors represent a burgeoning field of research featuring small devices capable of autonomous movement in liquid environments through catalytic reactions and/or external stimuli. This review delves into recent advancements in light-driven semiconductor-based micro/nanomotors (LDSM), focusing on optimized syntheses, enhanced motion mechanisms, and emerging applications in the environmental and biomedical domains. The survey commences with a theoretical introduction to micromotors and their propulsion mechanisms, followed by an exploration of commonly studied LDSM, emphasizing their advantages. Critical properties affecting propulsion, such as surface features, morphology, and size, are presented alongside discussions on external conditions related to light sources and intensity, which are crucial for optimizing the propulsion speed. Each property is accompanied by a theoretical background and conclusions drawn up to 2018. The review further investigates recent adaptations of LDSM, uncovering underlying mechanisms and associated benefits. A brief discussion is included on potential synergistic effects between different external conditions, aiming to enhance efficiency—a relatively underexplored topic. In conclusion, the review outlines emerging applications in biomedicine and environmental monitoring/remediation resulting from recent LDSM research, highlighting the growing significance of this field. The comprehensive exploration of LDSM advancements provides valuable insights for researchers and practitioners seeking to leverage these innovative micro/nanomotors in diverse applications.

## 1. Introduction

Micro/nanomotors are small devices capable of moving autonomously in a liquid medium using catalytic reactions or an external stimulus (chemical fuels, magnetic, ultrasound, or light). The main features generating the movement of these motors are morphological asymmetry (generally created by the deposition of metals on one side of the motors, creating two distinct hemispheres) and the consumption of chemical fuels, as presented later in more detail. Such small automotive devices were recognized for performing sophisticated tasks while moving in different liquid media, which allows them to be applied in the most diverse areas, namely detection, biomedicine, and environmental remediation [1]. This versatility has made these micro/nanomotors one of the most promising research topics worldwide.

Regarding the energy source used to generate propulsion, studies on micro/nanomotors can be grouped into five categories corresponding to the type of energy involved: chemical fuels [2,3], magnetic fields [4,5], electric fields [6], acoustic [7], and light [8,9]. Motors powered by chemical fuels often form bubbles or chemical gradients to move. The main challenge in this type of motor is movement control and toxicity. Considering that the chemical fuel most described in the literature is hydrogen peroxide (H_2_O_2_), the intrinsic toxicity may limit the applicability of these catalytic micro/nanomotors in the biomedical field. In this case, H_2_O_2_ can be catalytically decomposed into O_2_ and H_2_O molecules on the surface of micro/nanomotors, creating a chemical gradient that triggers uncontrolled autonomous movement. Ultrasonication, a recently tested energy source, provides robust propulsion but requires more sophisticated equipment. Magnetic motors have their movement controlled under adjusted magnetic fields, which limit their application to specific functions. Electric motors follow electrical gradients, limiting themselves mainly to two-dimensional movements. Motors powered by light, especially solar, have gained prominence due to their advantages, such as their low cost, ease of control, and ability to carry out complex tasks. Light, an abundant, remotely propagated, clean, and controllable energy source, is a promising option for driving micro/nanomotors [10]. The most recent reviews describing the main advances in the propulsion conditions of light-driven micro/nanomotors [9,11] or light-powered micro/nanomachines without the use of chemical fuel [8] to make the process greener and expand possible applications date back to 2018. Therefore, the present review focuses on results published in the past five years (2018–2023) on light-driven semiconductor-based micro/nanomotors (LDSM), exploring the advances in the synthesis, mechanisms of motion, and applications in the environmental and biomedicine fields. This review starts with a short theoretical introduction about micromotors and their propulsion mechanisms. Next, it discusses the most studied light-driven synthetic micromotors (LDSM) in the recent literature, highlighting their main advantages. The characteristics of LDSM that have the most significant impact on propulsion, such as surface features, morphology, and size, are then presented. External conditions related to the light source and its intensity are also discussed, for performing an important role in optimizing propulsion speed.

For each property, a brief theoretical introduction is provided, along with conclusions drawn up to the year 2018. Subsequently, the review explores the latest adaptations of LDSM, aiming to uncover the underlying mechanisms and their respective benefits. A short discussion is included on the potential synergistic effects between the different external conditions of LDSM, with the goal of enhancing efficiency—a topic of high importance, but one that has been relatively underexplored to date.

In conclusion, the review highlights the main applications that have emerged in recent years from LDSM research, particularly in the fields of biomedicine and environmental monitoring/remediation.

### Propulsion Mechanisms

For a better understanding of the subsequent sections, it is important to make a summary of the different propulsion mechanisms already described in the literature for light-driven micro/nanomotors.

Small devices moving through a fluid face a challenge due to their limited motion at a low Reynolds number [12]; that is, they encounter a powerful viscous force. Scientists are working on various methods to tackle this issue and enhance the devices’ controlled movement, countering the random Brownian motion that typically hinders them, as shown in Figure 1.

In photothermal propulsion, the thermophoretic force resulting from the temperature gradient drives the motor as the temperature decreases. When light intensity increases, it generally generates more heat. This increase in temperature creates a thermal gradient, that is, a temperature difference between the hottest and coldest regions. The micro/nanomotors are designed to respond to these temperature changes, as they are made of heat-sensitive materials that expand or contract with temperature variations. When there is a more significant thermal gradient (due to a higher light intensity), the response of these materials is more pronounced. This change in the shape or volume of heat-sensitive materials generates an acting force on the micro/nanomotors, causing them to move or perform some specific action. Thus, in short, temperature controls the speed of these motors through its effect on heat-sensitive materials, influencing the actuation force they generate.

For example, Yang et al. described the propulsion mechanism of micro/nanomotors irradiated with a near-infrared (NIR) light source, in which the absorbed electromagnetic energy was dissipated with heat. Then, a thermal gradient was formed around the surrounding liquid [13]. This increase in temperature gradient resulted in movement by the auto-thermophoresis of the photothermal materials. This way, it was possible to control the motors’ movement through the illumination intensity of the NIR laser. Greater light intensity led to a more significant temperature gradient, resulting in a greater actuation force of the micro/nanomotors.

In auto-electrophoretic propulsion, light plays a crucial role in generating an electric field that, in turn, drives micro- or nano-sized particles. This phenomenon is based on the principles of electrophoresis, which is the movement of charged particles in response to an electric field. In other words, light falls on the particles, activating specific properties that make them electrically charged. The presence of the same light creates an electric field around the charged particles due to electrophotonic interactions. The local electric field results from the asymmetric distribution of charged ions, which are generated from photocatalytic reactions on the surface of micro/nanomotors, specifically on the surfaces of different types of materials. As the electric field is generated under light irradiation, the charged micro/nanomotors are caused to move in response to the area [14,15]. In short, light acts as the trigger to generate electrical charges on particles, and the interaction of these charges with the light-induced electric field results in particle movement. However, it should be noted that depending on the properties of specific materials, such as the metals and semiconductors used to prepare light-driven micro/nanomotors, the type of reaction triggered by the incidence of light can be differentiated. This approach is explored in several applications, such as manufacturing light-driven micro/nanomotors for transporting cargo on a microscopic scale.

In bubble-induced propulsion, the force generated by the bubbles drives the motors in an opposite direction. In this case, light plays a key role in generating and manipulating gas bubbles. Light falls on a light-sensitive material, such as a photocatalyst, and the light-sensitive material undergoes photolysis (the process in which light breaks down precursor molecules, releasing gas (such as oxygen or hydrogen) as a byproduct). The released gas accumulates on the surface of the micromotor, forming bubbles. The accumulation of bubbles creates a pressure difference on the surface of the micromotor. This results in the expulsion of the bubbles, creating an impulse that drives the micromotor in the opposite direction.

In other words, as bubbles are released from the surface of the motors, resulting from the mostly redox reactions that occur, this unleashes a driving force away from the surface of the motors [16,17]. It is essential to highlight that in these cases, the driving force is assumed to be balanced with the viscous drag force if the vertical forces are ignored.

Finally, in auto-diffusiophoretic propulsion (also named osmotic propulsion), light is crucial in creating chemical gradients that lead to particle movement. The light falls on the surface of the particles, activating specific chemical reactions or surface interactions that change the local chemical composition. As a result of irradiation, a local chemical change occurs that leads to the formation of concentration gradients of certain substances around the particles. The resulting concentration gradients induce a directional movement of the particles, known as diffusiophoresis. This movement occurs due to diffusion forces arising when particles move toward lower- or higher-concentration regions. The directional movement of particles, induced by diffusiophoresis, results in autodiffusiophoretic propulsion.

In short, light creates chemical gradients around particles, triggering directional movement based on diffusion forces. This phenomenon is especially exploited in micromotors and micro/nanometer-scale particles to achieve controlled propulsion in liquid environments [18,19].

There are other movement mechanisms, less described in the LDSM literature, namely photoinduced deformation propulsion, which refers to a propulsion method occurring through shape change (bending, expanding, or contracting) activated by light (photoinduced). This mechanism is particularly associated with photoactive polymeric materials [20] and liquid crystals [21,22] and may also be referenced in metallic compounds that possess properties changing according to the specific light incidence (in terms of intensity, polarization, or frequency). In the case of LDSM, this movement mechanism can be associated with LDSM hybrids with polymers [23]. Light-induced surface tension variation is another movement mechanism that exploits the ability to alter the surface tension of liquids through exposure to light, in order to generate movement and force on LDSM. In this case, a semiconductor with chemical or physical properties altered by light exposure can directly affect the surface tension of the surrounding liquid, resulting in its movement [24]. Another example is localized light exposure, which can create gradients of surface tension, with areas of higher and lower tension. These gradients can induce liquid flows or move LDSM placed on the liquid surface, exploiting the phenomenon known as the Marangoni effect, where the liquid flows from areas of low surface tension to areas of high surface tension [25,26].

## 2. Light-Driven Semiconductor-Based Micro/Nanomotors

LDSM are mobile devices created from materials that conduct electricity activated by light, as is the case with some metal oxides (i.e., TiO_2_, ZnO, CuO_2_, BiVO_4_, WO_3_) [9]. Materials, in general, have two energy bands: the valence band (highest occupied) and the conduction band (lowest unoccupied), separated by an interval called band gap (*Eg*) that defines electrical conductivity. Non-conducting materials have a large *Eg* (≥4 eV), which limits electron transfer. Semiconductors have intermediate conductivity (*Eg* between 1 and 4 eV), and the higher the *Eg* value, the lower the range of radiation wavelengths capable of triggering propulsion. In other words, the higher the *Eg*, the lower the visible (Vis) light absorption capacity of the semiconductor. The compilation of recent data from the literature, presented in Table 1, shows that to produce LDSM, materials with photocatalytic activity, namely metal oxides (highlighting TiO_2_, CuO_2_, and ZnO), are mainly used in these materials. These are chosen because they can utilize energy from external light sources and surrounding chemicals to achieve efficient propulsion. Its photocatalytic properties convert optical and chemical energy into mechanical movement through photocatalytic reactions, making it possible to obtain motors with propulsion modulation through chemical concentration or light intensity. In other words, micro/nanomotors based on photocatalytic semiconductors can have controlled propulsion, for example, by simply regulating the intensity of the light source with on/off tests (turning on the irradiation source, letting it reach the maximum speed and turning off the source), or even by the position of the irradiation beam. Furthermore, they can operate with low optical and chemical energy input, making them ideal for industrial applications. Another important factor to consider in this type of motor is the production of free radicals during photocatalytic reactions that offer the potential for environmental remediation, especially in the degradation of organic pollutants, since the gradient or asymmetric distribution of photocatalytic products generated by photocatalysts when exposed to the light of a specific wavelength (such as ions, molecules or gases), trigger cycles of redox reactions with the surrounding reactants that interfere with propulsion, which may or may not be beneficial. 

Table 1 presents a selection of the articles published in the past 5 years (2018–2023) on LDSM having, in our view, the most significant implications on the optimization of light-driven movement. This table shows the main factors to consider for a more efficient and controlled propulsion speed: chemical constitution, morphology, scale, mechanisms of action, light source, and respective intensity. For each study, optimized movement speeds were presented to facilitate comparison.

### 2.1. Semiconductors Most Used in the Preparation of Light-Driven Micro/Nanomotors

As previously mentioned, and corroborated by Table 1, the semiconductors most used to prepare LDSM are metal oxides with photocatalytic activity and some carbon sources. In the following sub-sections, the semiconductors most used for preparing LDSM will be presented. After a summary of the previous state of the art, innovative and representative examples of the advances achieved in the past 5 years are presented.

#### 2.1.1. Light-Driven TiO_2_-Based Micro/Nanomotors

TiO_2_ has always been the most used semiconductor metal oxide to produce LDSM, due to its photocatalytic characteristics associated with chemical stability under light irradiation, low cost, and low toxicity [9]. Light-driven TiO_2_ micro/nanomotors have been widely reported, considering the high applicability of this semiconductor in photocatalysis. With TiO_2_ being the semiconductor most used to produce photocatalysts and LDSM being sophisticated versions of photocatalysts, extensive use of TiO_2_ in the production of LDSM is expected. The first example of TiO_2_-based micromotor systems was reported in 2010 by Sen and collaborators [62]. Since then, several studies have been reported with TiO_2_-based micro/nanomotors [15,16,18,28,31,33,36,39,41,42,43,63,64]. Many parameters have been scrutinized in recent years to prepare more efficient TiO_2_-based micromotors, namely the morphology, scale, and degree of crystallinity of the semiconductor. The spherical and tubular shape on the micrometric scale has been more widely described in the literature due to the simplicity of the synthesis methods, as well as the reproducibility and controllability of other important parameters [11]. However, in the recent past, attempts have been made to reduce the scale, a slow process considering the difficulty inherent in the reproducibility of the synthesis process. The structure asymmetry (Janus structures which will be explored in more detail in Section 3.1.3) is one of the most discussed parameters for obtaining propulsion for light-driven TiO_2_-based motors.

The agglomeration effects and propulsion conditioned by the interaction between motors were explored and often avoided through the crystallinity of the semiconductor [27]. The different degrees of crystallinity, easily obtained with this type of semiconductor (anatase and rutile), are also the subject of extensive study, considering the importance of this parameter for optimizing the photocatalytic process and consequent acceleration of the propulsion mode. The anatase phase of TiO_2_ has generally been considered the most effective crystalline phase for obtaining TiO_2_-based micro/nanomotors with an improved photocatalytic activity, resulting in faster movement. In other words, the main motivation in the study and development of this type of motor, until 2016, was to understand the propulsion mechanism inherent to the different morphologies and ways of optimizing the same mechanism to accelerate the light-driven movement. More recently, in addition to improved propulsion speed and testing less dense structures [27], the focus has been on overcoming one of the significant limitations of the use of TiO_2_ in LDSM, the high value of *Eg* (~3 eV), which restricts the activation of TiO_2_-based micromotors upon UV light irradiation (≤390 nm). Different strategies have been explored, in the past few years, to expand the light-absorption capabilities of TiO_2_-based micromotors to the visible region, i.e., a combination of TiO_2_ with a Vis-light-absorbing photocatalyst by a p-n junction [32,39,43], surface doping [29,31], dye sensitization [65], or reducing TiO_2_ to black TiO_2_ [66]. Gibbs and collaborators [43] recently explored joining two photocatalytic semiconductors, TiO_2_ and CuO_2_, to prepare hybrid micromotors. In this study, a p-type semiconductor (the dominant charge carriers are positive “holes”) such as CuO_2_ was conjugated with an n-type semiconductor (in which the dominant charge carriers are electrons) such as TiO_2_ to modify the electrical conductivity of the hybrid photocatalyst. These micromotors were produced by electron beam evaporation, using SiO_2_ microspheres as a support. Initially, TiO_2_ was deposited at an oblique angle on SiO_2_, then the sample stage was rotated by 180°, and Cu was deposited on the “tail” of TiO_2_, generating chevron-shaped morphologies (Figure 2a). These hybrid micromotors showed self-propulsion in water under UV and Vis irradiation (380–450 nm), through translational and rotational movements, respectively. The use of the n-type semiconductor (TiO_2_) resulted in the introduction of a small amount of an impurity into the CuO_2_ crystalline structure that modified its electronic properties, giving rise to additional energy states in the band gap, which thus reduced the band gap. As shown in Figure 2b, a potential barrier is formed at the interface of the CuO_2_ (p-type semiconductor) and TiO_2_ (n-type semiconductor) junction, creating an internal electric field. This interface acts as a barrier to prevent e^−^/h^+^ recombination, one of the strategies to increase the photocatalytic efficiency of TiO_2_-based motors. This strategy, in addition to being able to be applied to other semiconductors, such as ZnO, has the advantage of being a more economical approach since noble metals, widely used in the production of efficient micromotors [9], are not necessary for this approach.

Sensitization with organic dyes of Si/TiO_2_ nanotrees was demonstrated by Zheng et al. [65] to obtain Vis-light-driven nanomotors. In this study, dyes with a different sensitivity to Vis light (green, red, and blue) were incorporated into the surface of the Si/TiO_2_ nanotree to control the nanomotors’ movement depending on the source of irradiated light. The method of producing large-scale nanotree structures was based on a modified metal-assisted chemical etching process. This was followed by the hydrothermal growth of the TiO_2_ nanowires. Before the dye adsorption step, the Janus TiO_2_/Si nanotree forest was heat treated in air at 450 °C for 30 min. For sensitization, three different dyes were used—N719, D5, and SQ2. The prepared nanotree forests were stained by immersion in an ethanolic dye solution, for incorporation of the dyes, each with a Vis light absorption capacity in a particular range, as seen in Figure 3a. Thus, when blue or red Vis light is applied, as shown in Figure 3b, the groups of Si/TiO_2_ nanomotors with SQ2 and D5, respectively, will be activated and presented with a movement controlled by the action of the corresponding light source.

The reduction of TiO_2_ to its black form is a method that can decrease the bandgap energy. Black TiO_2_ can be obtained through thermal treatment (temperature ramps up to approximately 400 °C) in controlled environments, such as reducing atmospheres (e.g., nitrogen atmosphere) or under vacuum conditions. This process can induce changes in the crystal structure that can involving the introduction of defects, such as oxygen vacancies, interstitial defects, and structural disorder, thereby creating additional electronic states within the bandgap and reducing the bandgap energy.

Consequently, electrons can be excited by lower-energy photons, such as those from visible light, thereby increasing the absorption efficiency of TiO_2_ in this region of the electromagnetic spectrum. The utilization of black TiO_2_ to create LDSM based on TiO_2_ with propulsion driven by visible light has been discussed in the literature as a beneficial alternative for achieving solar-light-driven self-propulsion, particularly suitable for biomedical applications [66].

The nanomotors, specifically SQ2 (red) and D5 (blue), exhibited spontaneous trajectories forming the letters ‘r’ and ‘b’ as they navigated under the influence of blue and red light, respectively (Figure 3c). This happened because the red illuminations (660 nm) were constant in terms of position and irradiation intensity, while the 475 nm blue illuminations were lit sequentially from various directions to rotate the sensitized micromotor contained the D5 dye, regardless of the direction of the red illumination (Figure 3b). During this process, the orientation of the SQ2-loaded micromotor was locked by the illumination direction of the 660 nm red light and did not show any rotation with the blue light source. Comparatively, fixed blue 475 nm illumination with rotating red 660 nm illumination only drove the rotation of the sensitized micromotor SQ2 without affecting the alignment of the sensitized micromotor D5 (Figure 3c). This study demonstrated that dye-sensitized micromotors could indeed be controlled independently.

Therefore, this work had a double importance, considering obtaining a propulsion response driven by Vis light and the possibility of obtaining independent movement for each nanomotor to create an essential cooperative action for different applications, particularly in biomedicine.

#### 2.1.2. Light-Driven ZnO-Based Micro/Nanomotors

ZnO also stands out as a semiconductor with auspicious photocatalytic properties, boosted by its non-toxicity, high electron mobility, and an *Eg* ~3.37 eV. This characteristic gives ZnO a wide range of radiation absorption and high photostability, in addition to being a semiconductor that easily converts light energy into kinetic energy to drive motors. Furthermore, ZnO is known to have a high electron and hole mobility, which may contribute to a faster response to light stimuli. However, despite these advantages, its application in motor research is limited due to its high density and low colloidal stability. Even so, ZnO-based photocatalytic materials have been recognized as promising agents in the photocatalysis of environmental pollutants, with potential applications in light-driven micro/nanomotors.

The compilation of different studies carried out until 2018 with light-driven ZnO-based micro/nanomotors described several applications and developed different key points that include propulsion mechanisms, energy efficiency, and possible biomedical applications. Many studies have explored different propulsion mechanisms for ZnO-based micromotors, such as the release of oxygen bubbles or the generation of electrochemical currents under the influence of light, to make these motors faster.

There was an emphasis on optimizing energy efficiency, trying to maximize the conversion of light energy into the efficient movement of micromotors.

In the past 5 years, the focus in the study of ZnO LDSM has continued to be the optimization of the propulsion mechanism to accelerate the movement, but also the concern of its the directional control [45,46,47,48,50,67]. The ability to control light-driven ZnO-based micro/nanomotors remotely by playing with the positioning and intensity of the light source has been widely studied, aiming at the development of precise control systems. Combined with this, the exploration of the nanometer scale proved to be effective in improving propulsion both in speed and direction control. An example of this was the study of light-driven Au–ZnO nanorod motors developed using hydrothermal methods to remove antibiotics from aquatic environments [45]. This study considered different parameters, such as the size and diameter of the nanorods, to obtain nanomotors with controlled and efficient propulsion. The 0.85 ± 0.16 μm size combined with a diameter of 90 ± 24 nm were considered ideal parameters for improved movement. UV light intensity was also considered for propulsion control purposes. With on/off tests, it was possible to verify that when the light was turned off, the speed of the nanomotors gradually decreased from 24 ± 4 μm·s^−1^ to Brownian motion. The presence of Au allowed the optimization of the separation of charges on the nanorods’ surface, positively affecting both the self-propulsion and the photocatalytic degradation performance of tetracycline.

The study of the size of ZnO/Pt Janus motors (1.5 to 5 μm) showed a notable difference in light-driven speed resulting from different propulsion mechanisms due to different porosity scales. In other words, the smaller micromotors (1.5 μm) exhibited self-diffusiophoresis due to their microporous structure, obtained through the catalytic decomposition of H_2_O_2_ inside the particle. The rapid propulsion observed in the 5 μm micromotors was attributed to the mesoporous interface structure, leading to efficient bubble nucleation and expulsion through the generation of small, high-density microbubbles. The impressive bubble propulsion speed exhibited by the mesoporous ZnO/Pt micromotors set a new standard when compared to previously reported Janus micromotors utilizing non-heavy materials such as PS and silica. In this way, it was possible to obtain a very fast micromotor (350 μm·s^−1^) by adjusting the diameter of the Janus structure to 5 μm, representing a double acceleration compared to smaller micromotors [48].

Integration with other materials, especially through hybridization with other semiconductors, has also been highlighted in recent years, considering the need to overcome some limitations inherent to semiconductors in their individual state, through synergistic and/or cumulative effects. Some studies have explored the integration of ZnO with other materials to further improve the performance of micromotors. For example, Zhang and collaborators presented core-shell nanomotors utilizing Sb_2_Se_3_ nanowires as the core [47]. These nanowires were prepared via the physical vapor deposition method, with a polycrystalline ZnO shell (150 nm) applied to the Sb_2_Se_3_ nanowire through atomic layer deposition. The ZnO shell functions as a charge collector for both Sb_2_Se_3_ nanowire photovoltaics and nanomotors. This study showcased a breakthrough in micromotor navigation. A significantly faster and more precisely targeted navigation of the general micromotor was achieved through the strategic assembly of two cross-aligned nanomotors and the application of polarized light. This investigation focused on the polarization state of light, exploiting the anisotropic crystalline structure of the motor that facilitated the preferential adsorption of polarized light parallel to the Sb_2_Se_3_ nanowires.

As a result, the Sb_2_Se_3_/ZnO core-shell nanomotor exhibited robust dichroic swimming behavior, displaying a movement speed three times faster when illuminated with parallel polarized light compared to perpendicular polarized light. Moreover, incorporating two cross-aligned dichroic nanomotors led to the development of polarotactic nanomotors, endowed with the ability to control the movement direction based on the polarization of the incident light. This advancement not only enhances the speed and precision of micromotor navigation but also opens up possibilities for applications where directional control is crucial.

#### 2.1.3. Light-Driven CuO_2_-Based Micro/Nanomotors

CuO_2_ is also one of the photocatalytic semiconductors used for the production of light-driven micro/nanomotors, mainly due to the reduced *Eg* (approximately 0.75 eV) that makes the motors based on CuO_2_ Vis-light-responsive. The high surface area that increases the adsorptive capacity of these micro/nanomotors is also an important characteristic of this semiconductor.

Research into light-driven micromotors using CuO_2_ has been and continues to be less common than that using materials such as ZnO or TiO_2_. However, in general, the few publications that have been presented on light-driven CuO_2_-based motors have focused on optimizing the efficiency of converting light energy into efficient propulsion in specific applications such as biosensing and drug delivery [10], through tests with structures with different morphologies and sizes.

In the past 5 years, the number of publications on light-driven CuO_2_-based motors has increased, with the main objective of optimizing the process of converting light energy into movement. The main novelty involves the optimization of surface characteristics related to the crystalline state of the semiconductor and the capability to manipulate different crystalline facets on the surface of the same LDSM [53,54,55,60,67]. For example, using CuO_2_ as a semiconductor associated with asymmetric structures with specific geometry was one of the strategies described to achieve Vis-light-driven micro/nanomotors [55]. In this study, the authors considered CuO_2_ to prepare the micro/nanomotors because it is a semiconductor with a high surface area, responsive to Vis light irradiation, and with different crystalline phases that can enhance self-propulsion. The previous characteristics were combined with a truncated octahedron morphology that allowed the preservation of the controllable crystalline facets {100} and {111} in a single colloid. These morphological and crystallinity characteristics are essential to achieve directed and efficient propulsion. These parameters will be covered in greater detail in the following sections. In this case, it is necessary to highlight the relevant choice of CuO_2_ as a base material, which combined all these essential characteristics to prepare highly efficient Vis-light-driven micro/nanomotors.

Another work was described on Vis-light-driven micro/nanomotors based on CuO_2_, and in this case, the choice of semiconductor was made considering the possibility of introducing oxygen-vacancy-based Cu_2+1_O to improve the self-propulsion of the micro/nanomotors [53]. Oxygen vacancies are crystalline defects that trap photogenerated electrons, increasing the materials’ photocatalytic activity. These motors can be synthesized in a single step, with low costs and without additional modifications. Cu_2+1_O micro/nanomotors demonstrate excellent propulsion on biocompatible fuels such as tannic acid. These motors reached maximum speeds of up to 18 µm·s^−1^ in pure water under Vis light, comparable to conventional Pt-based catalytic micromotors powered by toxic H_2_O_2_ [68].

#### 2.1.4. Hybrid Light-Driven Semiconductor-Based Micro/Nanomotors

The mixture of two or more semiconductors in the same LDSM has the advantage of reconciling and/or creating synergistic effects between the main characteristics of each semiconductor, as well as overcoming some of the limitations of each semiconductor in its individual state. Since work with LDSM began, one of the limitations most mentioned by researchers has been the limited charge separation for light-driven micromotors of single-component semiconductors, which leads to weak propulsion power and low motion. Therefore, it is important to construct hybrid structures to increase the photocatalytic performance, improve the motor motility, and satisfy the demand for practical applications.

Chen et al. described the production of Vis-light-driven Cu_2_O@CdSe micromotors with excellent cationic dye removal capabilities [54]. The gap between the Cu_2_O and CdSe semiconductors formed a heterojunction that effectively inhibited the recombination of photogenerated e^−^/h^+^ pairs and improved the photocatalytic activity and, consequently, the propulsion speed of the Cu_2_O@CdSe micromotors. Self-propulsion reached a maximum speed of 42 μm·s^−1^ in biological environments due to the formation of the asymmetric chemical concentration gradient around the motors. Furthermore, the adsorbent capacity of the Cu_2_O@CdSe micromotors was confirmed with a fast adsorption rate of 96% in methyl blue after 10 min, mainly because of the high surface area of Cu_2_O.

Another process for obtaining Vis-light-driven hybrid nanomotors was proposed by Ge and collaborators [16]. This work described new nanomotors that moved in response to Vis light by Janus TiO_2_/MnO_2_ nanoparticles and aimed to create cost-effective nanomotors activated by Vis light without needing heavy metals for the required movement asymmetry. The synthesis process was based on the growth of MnO_2_ nanoflakes in one of the hemispheres of TiO_2_ spheres by the photoreduction of KMnO_4_ under aerobic conditions. In this case, propulsion was dependent on the use of H_2_O_2_. Considering the catalytic decomposition of H_2_O_2_ promoted by MnO_2_ nanoflakes, the engaging propulsion mechanism generates oxygen bubbles and consequently repels the nanomotors forward in the solution. For this reason, compared to other noble-metal-propelled Janus nanomotors (characterized by a self-electrophoretic propulsion mechanism), TiO_2_/MnO_2_ are still weak in propulsion speed (48 μm·s^−1^). To overcome this limitation, it was proposed to reduce the weight of the nanomotors or prepare a binary composite of a noble metal and MnO_2_ to further improve the catalytic performance in the decomposition of H_2_O_2_ and consequently improve propulsion.

Hybrid light-driven semiconductor-based micro/nanomotors is a recent topic that will certainly be emphasized in the future. The blend involving carbon constituents, specifically graphene, posited as a potential semiconductor, has proven to be a beneficial and exceedingly promising prospect

##### Light-Driven Graphene-Based Micro/Nanomotors

Graphene has exhibited promise as a semiconductor on a small scale in prior research, but the surge in studies pertaining to its application in LDSM has increased significantly. It is represented by a two-dimensional hybridized sp2-carbon sheet with attractive properties, such as a large surface area, good electrical conductivity, high intrinsic mobility, and excellent mechanical resistance [69]. Generally, it is used as a base material where catalyst nanoparticles are distributed asymmetrically to propel the graphene particle [44,52]. Using graphene as the main component provides new possibilities for applying light-driven micro/nanomotors, mainly in biomedicine and environmental remediation. For example, the coupling of graphene to micro/nanomotors makes it possible to create dynamic platforms for water decontamination by increasing the adsorption capacity of organic pollutants [70]. The first studies on light-driven graphene-based motors explored possible combinations of graphene with other materials to optimize the properties of micromotors (generally metal oxides) to increase the efficiency of converting light energy into movement. Another point addressed was modifications to the surface of the graphene-based motors to improve propulsion efficiency or to allow specific functionalization for specific applications. Despite all of this, the number of scientific publications on light-driven micromotors using graphene has been relatively limited, and it has been in the past 5 years that there has been greater interest in this material due to its unique properties [13,51,56].

The main points covered in the recent literature on light-driven graphene oxide (GO)-based micro/nanomotors have included using light to generate temperature gradients on the GO surface, resulting in directional movement [13]. This could involve absorbing light and converting that energy into motion, thus creating photothermal micromotors. For example, Yang and collaborators described the highly efficient light-driven micromotor composed of Cu_2_O@GO, capable of being propelled by various biocompatible fuels [13]. By the hybridization of Cu_2_O and GO, the micromotors exhibited an enhanced photocatalytic performance, enabling efficient propulsion under Vis and NIR light. Compared to conventional Cu_2_O micromotors driven solely by visible light, the Cu_2_O@GO micromotors achieved three times faster movement across the entire visible light spectrum when fueled by glucose. Moreover, the motors, benefiting from the enhanced photocatalysis and local photoinduced thermal effects induced by GO, can be powered by near-infrared light using biocompatible fuels like glucose, leucine, and urea solutions, reaching speeds of up to 11 µm·s^−1^ (for glucose). This innovative approach of improving light-driven micromotor performance through GO doping offers a promising avenue for developing micromotors with potential applications in biological environments.

The general interest in developing and optimizing methods to control graphene-based micromotor direction and speed using different light characteristics, such as intensity or polarization, has continued to be one of the main objects of study in the past 5 years. NIR light-steered graphene aerogel has been described as a promising micromotor with controlled self-propulsion [51]. These micromotors, with a Janus structure, were based on reduced graphene oxide aerogel microspheres (RGOAM) obtained by an electrospray methodology. Using aerogel particles, this methodology made it possible to obtain micromotors with crucial isotropic characteristics, namely low density, which allowed efficient self-propulsion without adding chemical fuels. The low density reduces the resistance to fluids on the water surface, making RGOAM motors faster (up to 18 µm·s^−1^) than the recent literature has reported. The direction and speed of movement in the water were controlled by the NIR light on/off effect. Considering the high adsorption and charge capacity provided by reduced graphene oxide, these micromotors were successfully applied in the active charge–transport–release of dyes on demand.

Yang et al. described a highly efficient light-driven Cu_2_O@GO micromotor that can be propelled by biocompatible fuels such as glucose [13]. In this study, the main objective of using GO was to obtain self-propulsion by Vis and/or NIR light. Compared to conventional Cu_2_O micromotors, which can only be driven by Vis light, Cu_2_O@GO micromotors showed speeds three times faster (17 μm·s^−1^) under the same conditions using glucose as fuel. The values were slightly lower than the propulsion speed achieved when using NIR light (11 µm·s^−1^). Proof of the propulsion by NIR light, although slower, associated with the use of biofuels could be crucial for potential biological applications.

It is important to note that the field of light-driven graphene-based micromotors has evolved rapidly, and a more significant number of publications is expected in the coming years aimed at possible specific applications, mainly in biomedicine. The exploration of other possible applications for these micromotors includes targeted drug delivery or micro/nanometric scale manipulation.

##### Light-Driven MXene-Based Micro/Nanomotors

Another class of 2D hybrid LDSM has emerged recently, utilizing derivatives of MXenes. MXenes are a group of 2D materials characterized by the general formula M_n+1_X_n_Tx (n = 1, 2, 3), where M represents transition metals such as Ti, X represents C and/or N, and Tx denotes the surface-terminating functionality, which can be −O, −F, or −OH. Their multilayer structure provides a large surface area, making them appealing for the development of new LDSM.

In a pioneering study, sandwich-like structures comprising TiO_2_@Ti_3_C_2_/Pt were created to serve as LDSM [71]. The Ti_3_C_2_ nanoflakes (derived from exfoliated MXene microparticles) were subjected to a Pt deposition process on one side, which allowed this side to be protected from oxidation while increasing the photocatalytic activity of the new LDSM. Simultaneously, the exposed side of the Ti_3_C_2_ nanoflakes spontaneously formed a superficial TiO_2_ layer through oxidation, triggering enhanced self-propulsion. This study presented an innovative proof of concept for fuel-free LDSM utilizing 2D materials, showcasing enhanced photodegradation capabilities against environmental pollutants.

Another investigation focused on converting Ti_3_C_2_Tx MXene microparticles into photocatalytic TiO_2_ through thermal annealing processes while preserving the distinctive multilayered structure of MXenes. Following Pt layer deposition and surface impregnation with magnetic γ-Fe_2_O_3_ nanoparticles, the resulting micromotors exhibited enhanced self-propulsion (16 μm·s^−1^) when exposed to UV light in pure water [72].

Following a similar line of research, Urso et al. [73] explored LDSM based on TiO_2_ microparticles derived from MXenes, utilizing Au and Ag, which revealed different electronic properties at the metal–TiO_2_ interface. After transforming Ti_3_C_2_Tx MXene particles into multilayered TiO_2_ through thermal annealing, thin layers of Au or Ag were asymmetrically deposited on their surface via sputtering. The resulting micromotors were tested in pure water with H_2_O_2_ as fuel. Under UV light, both types of micromotors exhibited self-propulsion, with Au-TiO_2_ micromotors demonstrating superior velocities compared to Ag–TiO_2_. The higher speed of Au–TiO_2_ micromotors was attributed to the stronger built-in electric field at the Au–TiO_2_ Schottky junction (a consequence of MXene derivation). This field enhances the separation of photogenerated e^+^/h^−^ pairs in TiO_2_ and promotes hole accumulation at the interface. In the presence of H_2_O_2_, micromotors deposited with different metals present different self-propulsion movements, highlighting the importance of metal–semiconductor interfaces (considering electronic properties) and the impact of the choice of metal on the performance of the micromotor. Au–TiO_2_ micromotors showed a slight enhancement in velocity when exposed to UV light, whereas Ag–TiO_2_ micromotors exhibited self-propelled motion even without UV light, driven by self-diffusiophoresis. When exposed to UV light, Ag–TiO_2_ micromotors demonstrated the highest propulsion speed due to the combined effect of Ag catalytic activity and self-electrophoresis.

## 3. Propulsion Optimization Parameters

LDSM propulsion depends on two major groups of factors: 1—characteristics of the motors (morphology, surface area, mesoporosity, dimensions, etc.); 2—conditions of the medium (type and source of irradiation, external stimuli, etc.).

Considering that most LDSM are obtained with photocatalytic materials, all factors that affect the photocatalytic performance of a material also consequently affect its propulsion capacity. In the following subsections, the main factors to be considered for efficient propulsion of LDSM will be explained, while the recent advances described in the literature will be emphasized.

### 3.1. Structural Parameters of LDSM That Affect Propulsion

#### 3.1.1. Scale

Scaling in LDSM can significantly influence the performance and applications of these devices. Microscaling in LDSM, especially in applications such as microfabrication or microengineering, can have several advantages. One of them is energy efficiency with a low energy required to obtain propulsion. Another advantage of microscaling is the possibility of a more flexible and personalized design, adapted to the specific needs of an application, which can result in more efficient and innovative solutions [9]. However, in nanoscale LDSM, the distance traveled by photogenerated charge carriers from the medium to the surface sites decreases, reducing the probability of charge recombination. Moreover, nanostructures necessitate a lower energy input for motion in contrast to their larger counterparts. Therefore, motors based on semiconductors with smaller particles are expected to present improved movement. A recent study on TiO_2_/Pt Janus submicromotors indicated that increasing the particle size from 0.8 to 7 µm caused an evident decrease in movement speed from 20 to 5 µm·s^−1^ [74]. The nanoscale is the one that has been the most popular in recent years, particularly in biomedicine, for controlled drug delivery and targeted therapy due to its ability to interact at the cellular level and move in more confined locations. Furthermore, nanometer-scale motors can efficiently harness light energy due to their higher surface area, resulting in a greater energy conversion efficiency. The fabrication of nanoscale devices generally requires less material compared to macroscale devices, resulting in lower resource consumption [75].

In other words, the main difference between the micro and nanometric scales is the surface area; therefore, the ideal size choice depends on its purpose. Another highlight in recent articles is the choice of scale considering a combination/synergism with other parameters, namely morphology, to make LDSM more efficient for a given application [76]. For example, Hormigos and collaborators [52] proposed a synthesis of motors based on graphene quantum dots. This study delved into nanometric tubular structures to assess the impact of reduced dimensions and a substantial surface-to-volume ratio, enhancing solubility properties and leading to improved synthesis yields and heightened reproducibility.

Furthermore, the benefit of condensing many functional groups into a small structure for different applications is known. In summary, this study highlights the significant impact of graphene material dimensions and structure on the preparation of self-propelled micromotors. The utilization of low-dimensional carbon nanomaterials (0D) demonstrated higher synthesis yields and reproducibility across batches compared to micromotors prepared from 2D GO. The enhanced solubility of 0D graphene quantum dots (GQDs), attributed to their small dimensions and increased functional group content, improved dispersion and electrodeposition efficiency.

#### 3.1.2. Morphology

Morphology is one of the most important aspects and transversal to all micro/nanomotors, for efficient propulsion. The morphology of LDSM can interfere with various properties and performance, directly influencing their behavior and effectiveness in different environments and applications. Since the beginning of LDSM, this has been the most studied parameter and one of the first to be optimized due to its high underlying impact. The main conclusions described in studies of different LDSM indicated that morphology can affect the propulsion efficiency, movement stability, motor interactions with the surrounding environment, load capacity, response to external stimuli, and durability of LDSM [9].

The shape and geometry of the micro/nanomotors directly affect the propulsion efficiency of the LDSM. More hydrodynamic surfaces and optimized shapes can result in more efficient travel when micromotors are in motion.

Morphology performs a crucial role in the directional stability of micro/nanomotors. Asymmetrical shapes favor a specific direction of movement, while symmetrical shapes can result in more unpredictable trajectories.

The interaction of LDSM with the environment around them, such as particles, obstacles, or surfaces, is influenced by morphology. The ability to avoid obstacles or selectively interact with certain materials may depend on the shape and structure of LDSM.

In light-responsive micro/nanomotors, morphology can influence sensitivity and the response to stimuli. Specific shapes can amplify or attenuate the response to certain signals.

The mechanical strength and durability of LDSM are affected by their morphology. More robust shapes and resilient materials can extend the life of micro/nanomotors in adverse conditions.

In summary, the morphology of micromotors is a critical design consideration as it can impact several properties that are fundamental to their effective performance in specific applications. In this review, particular emphasis will be placed on spherical and tubular shapes.


**Tubular shape**


Since the beginning of LDSM research, the tubular shape has been one of the most studied due to the advantages associated with its unique properties, such as high surface area, low density, and high mechanical resistance. Micro/nanotubes generally have a high surface area, providing more sites for photochemical interactions. This is beneficial in processes involving chemical reactions triggered by light. Furthermore, this type of structure is low-density and, at the same time, has a notable mechanical resistance. This is advantageous for building durable motors. Micro/nanotubes are known for their thermal and chemical stability, making them robust in various environments and operating conditions. All these characteristics make tubes promising materials for light-driven motors in diverse applications, from nanotechnology to biomedicine.

Micro/nanotubes have always been associated with the bubble propulsion mechanism [9]. In these typical cases, the tubular shape induces speed in the micro/nanomotors due to the efficient nucleation and ejection of bubbles. The chemical reactions that occur on the inner surface of the tube generate gas bubbles, which are released from the tube after growing into larger bubbles. Therefore, it is important to consider that the diameter of both ends of the tube should be slightly different, helping to release the bubbles [77]. During continuous bubble expulsion, the recoil force will drive the tube to move away from the final bubble release at high speed. Recent studies continue to explore the production of LDSM with a tubular shape and the bubble propulsion mechanism, giving special attention to the dimensions of the tubes and the respective diameters of the extremity [17,52,78,79]. For example, Wang et al. described the preparation of TiO_2_ nanotubes with self-propulsion controlled by UV light in the presence of H_2_O_2_, obtained by the electrochemical anodization process [17]. With the method chosen for preparing the tubes, the authors optimized the process to design tubes with a specific size and end dimensions adjusted to the best propulsion speed. In this work, tubular nanomotors were produced with a bubble propulsion mechanism. When the nanotubes were illuminated by UV light, the O_2_ molecules generated due to the photocatalytic decomposition of H_2_O_2_ grew into bubbles of different sizes, a consequence of confinement in the internal holes. The 2D nanomotor began to float over bubbles of different sizes, losing balance, which caused movement in the direction of large bubbles to smaller bubbles. Movement control was achieved by regulating the intensity of the light source.

Although the bubble propulsion mechanism is the most described in the literature, it the diffusiophoretic mechanism of tube propulsion has also been recently considered since, after lighting, the tubular micro/nanomotors also generate ions on the internal and external surface that are accumulated in more significant proportion on the outer surface compared to the inner cavity [52,79]. This gradient promotes a fluid flow through the tube’s interior, which pushes it forward, generating movement. The most recent literature has focused on carbon nanotubes, as they have a greater electrical conductivity, which facilitates the efficient transfer of electrical charges. It is a crucial function in systems that depend on converting light energy into electrical energy to drive motors. Furthermore, especially in quantum dots, the tubular shape is important as it enhances optical properties. In this case, the tubes can exhibit unique optical properties, such as light absorption at specific wavelengths. This makes it possible to tune the motors to respond to different light sources. Hormigos and collaborators [52] proposed a synthesis of graphene quantum dot-based motors with the shape of tubes. This combination allowed new nanomotors to be obtained with a greater surface area, consequently increasing the propulsion speed and improving the adsorption of Rh6G for future detection and environmental remediation.


**Spherical shape**


Analyzing the data in Table 1, it is possible to confirm that the spherical shape is the morphology most adopted in recent years for LDSM preparation. The use of spheres in LDSM presents specific advantages. Spheres can provide uniform charge distribution when functionalized with specific groups. This is useful in systems that efficiently transfer electrical charges to generate movement in response to light. The spherical shape can interact effectively with incident light from multiple directions, which can be exploited to optimize light energy capture. Furthermore, this shape allows it to roll and rotate quickly, which can be used to generate directional movement in response to light.

Of all geometries, the spherical shape is the simplest to obtain and functionalize. Generally, synthesis processes are more accessible, versatile, and economical, and upscaling is possible [80]. Spherical shapes are generally obtained by *O/W* microemulsion, hydrothermal and self-assembly methods from metal oxide precursors, or using spherical molds such as polystyrene or carbon spheres [81]. Spheres can be manufactured from various materials, including polymers, metals, and ceramics, allowing the selection of suitable materials for specific applications.

Depending on the chosen material, the spheres can be adapted to be stable in different environments, such as biological or extreme conditions.

The size and propulsion mechanism have always been the main studied parameters in LDSM spherical shape and continue to be highlighted in the recent literature to improve propulsion. In the past 5 years, the focus has been on obtaining LDSM without adding fuels for propulsion [8,46,82,83]. To this end, spherical micro/nanomotors offer a wider range of advantages, including the ability to move with appreciable speed at a low light intensity in pure water without surfactants and the ease of fabrication and modeling due to their simpler geometries [84]. Therefore, these types of devices offer a wider range of applications in the environmental domain, biomedicine, and analytical chemistry [9]. He et al. described light-driven ZnO/Pt spherical micro/nanomotors with the ability to align themselves to the direction of illumination, reaching speeds of up to 32 µm·s^−1^ without the addition of chemical fuels [46]. Pt deposition was crucial for obtaining the asymmetric Janus structure that confers the ability for phototactic movement. The movement resulted from the electrophoretic rotation induced by the asymmetric distribution of the zeta potential in the two hemispheres of the spherical micro/nanomotor in alignment with the electric field. Light-driven micro/nanomotors, based on other semiconductors such as TiO_2_/Pt and CdS/Pt, were also confirmed, indicating that dipole moment-induced propulsion can be applied to both autoelectrophoretic and diffusioelectrophoretic micro/nanomotors [46].


**Nanowires and nanorods**


Nanowires and nanorods are less frequent structures in LDSM. Their low use is due to some disadvantages. One of them is the manufacturing complexity requiring advanced nanofabrication techniques that tend to increase production costs and make large-scale implementation difficult. Nanowires and nanorods can be sensitive to the environment around them, including temperature variations and certain chemicals. This sensitivity can limit its stability and effectiveness under different conditions. Functionalization of these structures with specific groups can be challenging, and the lack of adequate functionalization can restrict the ability to carry out specific reactions or interactions necessary for motor function. Furthermore, when associating this morphology with the nanoscale, mechanical fragility can occur, as well as limitations in the load transported. In other words, nanowires and nanorods present a greater propensity for damage or breakage under the influence of mechanical forces, indicating vulnerability in this aspect. Limitations on the load transported suggest a consequence of this mechanical fragility as it restricts the amount or type of load that the structure can support due to its small size [85].

Despite all this, some studies have recently described the use of nanowires and nanorods morphology, as shown in Table 1 [29,45,47,49]. Ji et al. describe Au/Fe_2_O_3_ nanomotors with propulsion activated by Vis light and with a propulsion mechanism based on self-electrophoresis, a consequence of the shape and size (nanorods) [49]. The incidence of Vis light with a greater intensity (33 mW.mm^2^) and shorter wavelength (less than 552.2 nm), especially in the semiconductor part of the nanorods, generated a more evident distribution of electrons (diffusion and migration of protons), resulting in faster propulsion (30.7 µm·s^−1^). Sb_2_Se_3_/ZnO nanowires were described for drug delivery potential, considering the effective control of these motors to the type of light used to stimulate movement [47]. Due to the nanowire-shaped crystalline structure, these Sb_2_Se_3_/ZnO nanowires exhibit the preferential absorption of light polarized parallel to the nanowires, resulting in a propulsion speed three times faster under parallel polarized light than perpendicular polarized light. This proof-of-concept study demonstrated the importance of the nanowire structure associated with other surface characteristics for optimizing self-propulsion and obtaining more efficient micro/nanomotors. However, in addition to the structure, the surface characteristics, namely crystallinity, mesoporosity, surface area, and roughness, among others, are characteristics widely studied, in their individualized form or in combination, to improve the propulsion/activity of recently described micromotors.

#### 3.1.3. Asymmetric Structure (Janus Structure)

Creating asymmetrical structures, generally associated with the metal deposition and superficial alteration of one side of the structure, creating two hemispheres, is one of the most important parameters for improving self-propulsion. These asymmetric structures are called Janus structures and are closely associated with the spherical shape, although it should be noted that the Janus structure can also be applied to two-dimensional objects, such as microplates, to accelerate them [86,87,88]. These structures are known for their asymmetry in morphology/shape and are generally associated with photocatalytic spheres. In these cases, the asymmetry is generated with the deposition of a metal in one of the hemispheres of the photocatalyst, creating a tail effect, as represented in Figure 4.

It was Gennes who first used the term “Janus” during his Nobel lecture in 1991 to describe particles that show two hemispheres with different chemical and physical micro/nanostructures [84]. These structures have one hemisphere responsible for propulsion and the other responsible for triggering the activity, from drug release to acting as a cleaning agent for water purification [89]. The propulsion hemisphere can generally catalyze chemical reactions in the surrounding solution, which triggers self-propulsion through the specific mechanisms already described in the introduction.

The first studies on motors with an asymmetric structure focused on obtaining directed propulsion. This asymmetry was achieved by functionalizing one half of the particle with materials that interact differently with the environment. Initial studies focused on investigating these different materials, with noble metals (such as Au and Pt) being the most widely used. The choice of noble metals for Janus structures in LDSM offers a combination of efficiency, precise control, and versatility, making them ideal candidates for diverse technological and scientific applications. Due to the intrinsic characteristics of noble metals, they remain prominent choices in LDSM [18,27,30]. These metals are notable for allowing control of the speed and direction of Janus micromotors, as they favor the efficient conversion of light into energy. Furthermore, noble metals are known for their chemical stability, which is advantageous in different environments. This stability contributes to the durability and long-term effectiveness of micromotors. Another advantage of using noble metals is their ability to interact efficiently with different wavelengths of light, making them ideal for LDSM applications. This allows for more precise control over the operation of the micro/nanomotors in response to variations in incident light.

Regarding the recent literature, the use of noble metals for deposition in motors maintains its predominance, but the possibility of functionalizing the surfaces created with these metals is emphasized, allowing the connection of specific molecules or groups that can further modulate the properties and functions of micromotors.

New metals have been studied more recently, mainly to reduce production costs and add other properties to motors, such as using Co which provides magnetic activity [32].

A recent consideration in the optimization of Janus structures is the selection of metals beyond the traditional noble metals for deposition, considering not only the intrinsic characteristics of these metals but also other parameters relevant to the motors under development, such as the desired morphology and the light source to be used. Maric and collaborators [40] described different Metal/TiO_2_ Janus microspheres prepared with the deposition of diverse metals (Pt, Au, Cu, Fe, and Ag) tested to evaluate the propulsion characteristics as a function of UV light. Since these metals have different chemical potentials and catalytic effects for the water-splitting reaction, the propulsion speeds for each one in water without adding chemical fuels were determined. The Pt/TiO_2_ micromotors stood out with a propulsion speed equal to 8.9 µm·s^−1^, while the lowest speed was achieved with the Au/TiO_2_ micromotors (2.9 µm·s^−1^). It can be concluded that the speed of Janus micromotors strongly depends on the metal deposited on the microsphere’s surface and that the effective speed results from the synergistic effect of chemical and catalysis potential.

Likewise, Wang and collaborators [36] described tests to evaluate the propulsion capacity of TiO_2_ Janus micromotors with Cu and Au deposition in the presence or absence of a chemical fuel. The results indicated different behaviors and directions of movement under three different conditions (water + UV, H_2_O_2_ + Vis, and H_2_O_2_ + UV). The deposition of Cu in H_2_O_2_ resulted in a semiconductor CuO_2_ with photocatalytic properties, explaining the activity of Cu@TiO_2_ in dilute peroxide under Vis illumination, in contrast to non-oxidized Au@TiO_2,_ which was found to be inactive under these conditions. The Cu deposition also provided a faster movement by UV light in both hemispheres, allowing photocatalytic reactions and increasing the speed of the Cu@TiO_2_ Janus micromotors (38 µm·s^−1^) compared to Au@TiO_2_ (23 µm·s^−1^). Under water + UV conditions, both Cu@TiO_2_ and Au@TiO_2_ micromotors showed light-driven movement. An agglomeration effect was observed almost immediately upon motion in Au micromotors, while in Cu motors, this aggregation was much slower. These results are due to the intrinsic characteristics of the metals chosen to obtain the Janus structures. Cu undergoes semiconducting transformation in the presence of H_2_O_2_, resulting in the formation of copper oxide. This oxide exhibits photocatalytic properties, providing a rationale for the enhanced activity of Cu@TiO_2_ in diluted peroxide under Vis light, in contrast to the inactivity of unoxidized Au@TiO_2_ under similar conditions. Additionally, introducing H_2_O_2_ under UV light induces photocatalytic reactions in both hemispheres, leading to an accelerated propulsion speed of Cu@TiO_2_ motors compared to Au@TiO_2_.

The combination of two noble metals to form Janus structures has also been documented in the literature as being advantageous for creating the ability of a single micro/nanomotor to generate switchable propulsion modes and reverse the direction of movement depending on the stimulus exposed. In their work, Chen et al. [18], described the synthesis of Janus micromotors using TiO_2_ spheres obtained by Pt deposition with an external layer of Au. As observed in typical Janus structures, chemical reactions triggered by chemical or light stimuli were identified on the different surfaces of the micromotor. These reactions generate opposing propulsion forces on the Pt catalytic and TiO_2_ photocatalytic sides. An intermediate Au layer was introduced under the Pt surface to manage the switchable propulsions. This layer started to control the type of chemical propulsion, adjusting the photocatalytic reactions on the TiO_2_ side in response to the chemical reactions occurring on the Pt side. In this way, it was possible to establish a competition between the two reactions on opposite sides of the Janus microsphere in a single micromotor. This competitive effect was expertly controlled through the Au layer, adjusting the motors’ overall propulsion and direction of movement according to the dominant force.

A recent innovation was the deposition of metallic oxides to prepare the “tail” of the motors, which allows researchers to direct and improve the speed of new light-driven motors made up of non-semiconductor compounds such as SiO_2_. Some studies [34,35] highlighted the deposited TiO_2_ to create asymmetry and to give the motor photocatalytic capacity. In other words, these motors obtain a photocatalytic activity attributed to the TiO_2_ tail, which provides the chemical gradient necessary for the movement of the motor, as we can see in the scheme shown in the Figure 5.

Nicholls et al. [34] investigated the dynamics of photoactive micromotors with spherical Janus structures with distinct morphological variations. In this work, SiO_2_ spheres were used as the basis of the micromotor. TiO_2_ deposition was carried out using the dynamic physical vapor deposition method, which controlled the extent/direction of TiO_2_ deposition, giving rise to different motors with distinct “tails” that influenced the propulsion. After tests to determine the speed of movement of motors driven with UV light, it was possible to conclude that both the length of the photocatalytic tail and the diameter of the SiO_2_ spheres significantly affect propulsion in terms of speed and direction. Micromotors with tails (3.8 µm) had a propulsion speed three times higher (5 ± 2 µm·s^−1^) than analogs without tails. In continuation of this work, Holterhoff and collaborators [35] investigated the effects of interactions/proximity between Janus micromotors with elongated tails during self-propulsion. For this, the movement of micromotors with different tails on different SiO_2_ and Au surfaces was evaluated to study the effect of speed changes when the material is kept constant, but the surface roughness is changed to mimic the one that occurs with the proximity between moving micromotors. This study considered the surface-dependent osmotic flow in the electrical double layer above the stationary solid, which connects back to the nearby micromotor, consequently affecting its overall dynamics. In other words, the surface of micromotors affects not only the movement of the micromotor itself but also the movement of the underlying micromotor due to its proximity. Similar to what was previously described [35], it was found that lengthening the tail increased the speed of movement by approximately six times. Furthermore, it was found that the calculated speeds were different for micromotors with elongated tails and other surfaces ($\overset{–}{V}$_E_,_SiO2_ = 22 ± 5 µm·s^−1^, $\overset{–}{V}$*_E_,*_Au_ = 27 ± 9 µm·s^−1^). The Au surface benefited the movement of adjacent surfaces by approximately 25% compared to the SiO_2_ surface. This is because the UV light possibly reflects more on the Au, providing a greater light intensity and consequently higher speeds. Different Au thicknesses were tested (2–25 nm); in any thickness, the velocity values were similar and higher than those obtained with the SiO_2_ surface. These results confirm that the surface properties interfere with the movement, more precisely, the constitution of the surface, regardless of its thickness.

More recent studies have once again focused on the importance of the scale associated with the chosen Janus structure for preparing LDSM with enhanced propulsion. What has been discussed is that in large Janus motors (above 3 µm), bubble propulsion is standard, where visible gas bubbles are released from the active side [33]. In smaller Janus motors, the predominant mechanisms are diffusion and electrophoresis, converting mainly chemical energy into mobility [34,38,50]. For example, ZnO/Pt Janus nanomotors feature an electric dipole caused by an uneven distribution of zeta potential, which induces an electrophoretic rotation to suppress Brownian rotational diffusion. The rotation reaches equilibrium when the dipole aligns with the light, resulting in resistant directional motion. However, if the size of the motors is in the order of micrometers, due to the rapid rotation, achieving this specific directional movement is much more challenging, in addition to the fact that these micromotors generally exhibit greater diffusion when using chemical fuels [50].

#### 3.1.4. Crystallinity

Crystallinity is a characteristic widely studied in photocatalytic materials, as it is an important factor that strongly influences their photoactivity. A high degree of crystallinity involves fewer defects, which generally act as capture and recombination centers between photogenerated charge carriers. This functionality is associated with increased photocatalytic activity. Furthermore, the crystalline structure of LDSM components can influence their response to light. Crystalline materials can have specific optical characteristics that affect the absorption and conversion of light energy into kinetic energy. Consequently, everything that positively affects photoactivity also interferes with the self-propulsion capacity.

Crystallinity may also play a role in the stability and durability of LDSM. This condition is attributed to the superior thermal stability properties [90] as they exhibit high thermal conductivity due to the ordered arrangement of their atoms. This facilitates efficient heat dissipation, contributing to the thermal resistance of micromotors. The robustness gained from the crystalline organization minimizes the structural defects that increase the device’s durability. The crystalline structure provides superior resistance against mechanical deformations, ensuring the stability and longevity of the micromotor even in adverse conditions. Additionally, the crystalline nature imparts a superior hardness and resistance to wear, making the micromotors more resilient to abrasive processes [91]. Therefore, materials with a robust crystalline structure are expected to resist harsh environmental conditions better and extend the life of micromotors. This parameter was initially explored to influence the molecular orientation of materials in LDSM, to improve the responsiveness to specific light stimuli and the direction of movement. Giudicatti et al. [92] described rolled TiO_2_ tubes that, after annealing at 400 °C (a temperature widely studied to obtain TiO_2_ in the anatase phase), exhibited a propulsion speed of 60 µm·s^−1^ under UV light. Annealing temperatures ≥ 600 °C damaged the integrity of its tubular structure and were therefore not considered. Motors without heat treatment (called amorphous structures) demonstrated no movement. This happens because amorphous structures present an irregularity in the TiO_2_ network that does not occur with crystalline structures. In a crystalline material, the atoms are arranged in an orderly manner, creating well-defined energy bands. These bands facilitate the mobility of charge carriers, allowing them to move more freely through the material. However, when a material is amorphous or lacks crystallinity, the energy bands can become less defined, which can lead to a lower charge transfer efficiency.

In the context of an LDSM, the mobility of charge carriers is essential for generating the driving force. If the TiO_2_ used in the motor does not have a suitable crystalline structure, the conduction ability of the charge carriers may be impaired, resulting in the reduced performance of the light-driven motor, associated with the reduced photocatalytic activity of TiO_2_, resulting in a lower generation of photogenerated reactive species. Consequently, the lack of sufficient gradients on the surface of the micro/nanomotor impedes its movement.

More recently, crystallinity modulation has been explored as a strategy to tune the surface properties of LDSM. For example, introducing controlled facets into the crystal structure can be exploited to modify specific properties. Different crystalline facets in the same material may also play a role in the movement ability of a light-driven micro/nanomotor. Liu et al. [55] proposed a methodology for obtaining Vis-light-driven CuO_2_ nanomotors with efficient movement associated with crystallinity and crystalline facets exposed to the surface. The truncated octahedron shape was also important as it allowed the surface of the Cu_2_O nanomotors to maintain the controllable index crystal facets of {100} and {111} in a single colloid. Considering the exposed crystalline facets’ high crystallinity and distinct activity, a surface heterojunction was formed between the {100} and {111} facets to increase the recombination e^−^/h^+^. This condition allowed the octahedral Cu_2_O nanomotors to achieve autonomous and vigorous movement with biocompatible fuels under Vis light. These Cu_2_O nanomotors achieved a propulsion speed in water twice as fast (~11.0 µm·s^−1^) as that of polycrystalline spherical motors with low crystallinity. Therefore, this study demonstrated the importance of the crystallinity of the structures obtained for controlled and efficient propulsion.

#### 3.1.5. Density

Density is a very recent study parameter in LDSM. Although density is not often highlighted explicitly, older studies may have considered this parameter concerning the design, performance, and applicability of LDSM. The lack of interest in this characteristic of LDSM possibly had to do with its indirect influence on other motors’ properties. In other words, its action alone was not considered a possible improvement in the efficiency of LDSM unless associated with other properties of these motors [9].

However, in recent years, literature has already focused on density studies in LDSM, particularly in contexts where density can be considered when evaluating the power of micro/nanomotors, especially if limited energy sources are powering them [27,42,59,93]. For example, in LDSM that transports payloads, the density of the materials can influence the load capacity and efficiency in transporting these payloads. Density can also be a factor in the mechanical strength of micro/nanomotors. Denser materials can have a more robust structure, affecting durability and the ability to withstand adverse conditions.

The most recent studies on this parameter have considered the inherent advantage of reducing the density of LDSM to influence their ability to float instead of sink in liquid media (in the case of denser motors). This may be relevant in applications where vertical mobility is important. Furthermore, considering the balance between buoyancy and propulsion forces, the density of the materials used to construct LDSM can influence their mobility in water or other fluids. Janus hollow TiO_2_/Au micromotors were described to obtain improved propulsion through density reduction in the micromotors for possible application sin the active photocatalytic degradation of methylene blue [27]. The hollow spherical structures achieved a movement three times more accelerated than analogous solid structures (from approximately 15 to 35 µm·s^−1^). In this case study, the results obtained were justified by the reduction in density and the increase in surface area resulting from the hollow structure. In a similar study, hollow shell TiO_2_ microspheres [42] were tested as micromotors with multimodal movement by changing the external stimulus. This study tested different light intensity conditions, directions, and chemical fuel concentrations to enhance different movement responses in a single micromotor. The inherent internal mass asymmetry, caused by the hollow effect of the spheres, generated a spontaneously multimodal movement behavior between random Brownian propulsion (stochastic walking), negative phototaxis (movement against incident light), and negative photogravitaxis (motion against gravity) associated with the tested conditions. This study proves the need to relate different conditions and study synergistic effects for better propulsion results.

### 3.2. Surface Parameters of LDSM That Affect Propulsion

#### 3.2.1. Surface Area

From the beginning, the scientific literature on LDSM has addressed the surface area as a critical factor influencing the properties and performance of these devices. It has been demonstrated that the surface area of materials used in LDSM plays a crucial role in the efficiency of converting light energy into motion. Larger surfaces can offer more sites for adsorption (an important step for the photocatalytic process), increasing the overall efficiency of the process. The surface area can influence the stability of the photocatalytic response of micromotors. Larger surfaces can provide greater stability over time, contributing to more consistent operation. Furthermore, the surface area has also been considered relevant for the interactions of micromotors with the surrounding environment [1]. In biological or environmental applications, an optimized surface area can improve the ability to interact and respond to environmental stimuli. The ability of LDSMs to carry and deliver payloads can also be adjusted with a larger surface area. Larger surfaces can allow for a more significant loading of substances, increasing effectiveness in targeted delivery applications [94].

The most recent developments regarding this feature refer to optimizing the available surface area, in other words, the creation of higher surface areas considering new methods such as the creation of meso/macropores or rougher surfaces. Recent literature has indicated that a greater optimized surface area can affect sensitivity and the response to light stimuli through these conditions, making LDSM more efficient. For instance, greater roughness results in an enlargement of the surface area, representing a greater scope for chemical reactions, which, in turn, intensifies the absorbent and propulsive activity. A notable example is recent research carried out by the group of Wittmann and collaborators, which explored the influence of smooth and rough surfaces on the semiconductor/metal interface of a Janus nanomotor with movement activated by UV and Vis light [32]. As part of this study, two cobalt species were deposited on TiO_2_ nanospheres using different deposition methods, as represented in Figure 6A. This resulted in obtaining platelets@TiO_2_ structures (rough surface) by a dip coating method and cubes@TiO_2_ structures (smooth surface) by the Langmuir Blodgett method. A notable distinction was evident when comparing the propulsion behaviors of the particles obtained, as can be seen in Figure 6B–D. When tracking individual particles under UV light, cubes@TiO_2_ exhibit enhanced Brownian diffusion, possibly due to the introduced asymmetry combined with the photocatalytic properties of TiO_2_ as a peroxide degradation catalyst. On the other hand, rougher Janus particles (platelets@TiO_2_) showed rapid and diffuse movement under UV light (Figure 6B,D). This difference was more evident when blue light was used (Figure 6C,E). The platelets@TiO_2_ exhibited a propulsion velocity consistently greater than 13.9 µm·s^−1^, regardless of the irradiation source (UV or Vis). These values stand out significantly compared to the cubes@TiO_2_ analogs, which did not exceed 10 µm·s^−1^. According to the authors, cubes@TiO_2_ particles behaved similarly to uncoated particles, indicating that the reaction does not occur in different hemispheres and, therefore, does not result in stabilization.

#### 3.2.2. Porosity

Porosity is an important factor that can influence various properties and the performance of LDSM. In addition to being able to provide a greater surface area, porous structures affect the ability of LDSM to allow the efficient diffusion of substances, such as reagents and chemicals involved in photocatalytic processes, interfering with the penetration capacity of molecules and/or incident light.

The porosity of LDSMs can influence their ability to carry and deliver payloads. Porous structures can efficiently incorporate substances, improving targeted delivery applications.

Interactions with the environment can also be affected depending on the degree of porosity of LDSM. In biological or environmental contexts, porosity can affect the interaction with cells, tissues, or other substances present in the environment.

Another important point regards the stability and durability of LDSM, since porosity can offer a greater mechanical resistance and consequent stability in challenging environments.

More recent studies have focused on the porosity of LDSM as a factor that can impact their sensitivity and response to external stimuli, such as light [9,44]. The presence of pores can modulate the material’s sensitivity to incident light and thus create more efficient self-propulsion. Another focus has been creating highly porous LDSM to increase surface area but also make the motors less dense. With the aim of developing micromotors that move autonomously, without depending on chemical fuels, the integration of highly ordered and porous materials with an extensive surface area was considered a promising strategy to enhance the self-propelled speed and adsorption capacity of these colloidal motors [44]. Ikram and collaborators presented Janus colloidal light-driven micromotors, composed of a highly efficient organic metallic structure, which presented self-propulsion without the need for the presence of chemical fuels just by activation with UV and Vis light. Within the scope of this study, a functionalized metal organic framework (MOF) was used, which exhibits a hierarchical morphology with a regular porous network with a notable surface area (1741 m^2^/g). The propulsion speed values achieved were 40 and 20 µm·s^−1^ under UV and Vis light illumination, respectively, which, according to the authors, were a consequence of the high porosity of the MOFs. In another study carried out with mesoporous ZnO/Pt Janus micromotors, it was observed that the porosity combined with the roughness of these structures played a crucial role in their movement capacity without depending on any fuel [50]. Two types of ZnO-based particles were synthesized (see Figure 7), each with distinct surface morphologies and pore structures, achieved through the self-aggregation of primary nanoparticles and nanosheets (smooth and rough).

This study aimed to evaluate the effects of the interface, surface porosity, and facets in charge transfer responsible for the movement of mesoporous ZnO/Pt Janus micromotors. The results indicated that rough micromotors only showed movement with the addition of a chemical fuel. At the same time, in smooth motors, it was possible to obtain propulsion through autodiffusiophoresis without additional fuel. That is, roughness negatively impacted self-propulsion ability, which is explained by the strong interface between the semiconductor and the ZnO/Pt metal on the smooth surfaces, which provided fuel-free, light-driven propulsion. In other words, Pt allowed the masking of defects on the ZnO surface arising from hydroxide species and oxygen vacancies at crystalline boundaries, improving charge transfer at the interfaces and minimizing e^−^/h^+^ recombination rates. In this way, it was possible to conclude that porosity and surface roughness, in synergism, were important characteristics for obtaining micromotors based on ZnO driven by UV light without depending on chemical fuel. These features have been highlighted as relevant for other photocatalytic/metal heterojunctions with similar optical properties, such as TiO_2_/Pt and TiO_2_/Au. However, considering the light-activated plasmonic effect of the Au element (optical phenomenon related to the special interaction between light and free electrons on the metal surface), which can alter the previously discussed effects, is important. In other words, gold has surface plasmons, which undergo collective oscillations of free electrons on the metal surface when excited by light. These plasmons can amplify and interact with incident light in unique ways, potentially altering the propulsion behavior of the Au-motors.

#### 3.2.3. Functionalization

Early on, functionalization was considered a crucial strategy to modify and improve several properties of these devices. The main considerations in the initial studies focused on improving motor and application efficiency. For example, the functionalization of LDSM surfaces with specific materials such as metals or carbon sources can improve photocatalytic efficiency. Introducing functional groups or nanoparticles onto the surface can facilitate photocatalytic reactions and improve light absorption [9].

Functionalization can give LDSM the ability to respond to specific external stimuli. For example, introducing light-sensitive clusters could enable directional control with a specific light. The stability and durability of LDSM can also be adjusted through functionalization. Functionalized materials can be protected against degradation or corrosion under certain environmental conditions. Furthermore, introducing functional groups can modify the mechanical properties of micro/nanomotors, affecting their resistance and flexibility. This can be crucial in dynamic environments. Functionalizing with biomolecules in biomedical contexts can make LDSM more biocompatible and suitable for applications such as targeted drug delivery [11].

Recent studies continue to use functionalization mainly to optimize LDSM for specific applications. For example, hydrogel-based Janus micromotors functionalized with different nanoparticles have been described in the literature for applications in biomedicine and environmental remediation [33]. The manufacturing method for these Janus micromotors involved the encapsulation of hydrogel microspheres prepared in droplet microfluidics through a polymerization method. Then, a layer of functional nanoparticles of MnO_2_ (H_2_O_2_ catalyst), TiO_2_ (photocatalyst), and Fe_3_O_4_ (magnetic orientation) was added to the hydrogel spheres, resulting in Janus micromotors with different functionalities. Covering the micromotors with MnO_2_ triggered a new catalytic performance that, associated with the presence of H_2_O_2_, generated propulsion dependent on the amount of fuel used (0% wt of H_2_O_2_ induced Brownian movement, while in the presence of 3% wt of H_2_O_2_ the speed of propulsion characterized by the bubble mechanism was 135 µm·s^−1^). In the case of micromotors covered with Fe_3_O_4_, applying an external magnetic field to obtain movement was necessary since the effect of light was practically null. The Janus TiO_2_ micromotors presented a self-electrophoretic propulsion mechanism controlled by the intensity of UV light. After turning the irradiation source on and off, the speed decreased approximately four times until it reached Brownian motion (0.1 mm·s^−1^). These micromotors degraded 95.64% of the initial methylene blue concentration after 1 h. This work proved that the functionalization of the surface of micromotors affects the final activity, not only due to the possibility of creating new applications but also in optimizing the propulsion mechanism obtained.

Something relatively recent is the addition of specific functional groups to the surface that can give LDSM chemical selectivity, allowing preferential interaction with specific compounds. This could be relevant in several applications ranging from drug delivery, where selectivity is desired, to selective photocatalysis. There are different ways of obtaining selective photocatalysis [81], one of which corresponds to photocatalytic selectivity for the substrate. In this condition, the advantage is associated with the selective recognition of a target so that its selective adsorption occurs (dominant step of the process) and the selective degradation of the substrate occurs. One possible route for recognition on the surface of LDSM is through the molecular imprinting technique. This technique allows the creation of selective active sites capable of recognizing a target molecule. In this technique, the rigid three-dimensional structure is synthesized around a model molecule (template), subsequently originating cavities with high affinity for the template. In the background, a “memory” is thus imprinted during the material production phase, which exhibits complementary cavities in the shape, size, and position of the functional groups that interact with the template, creating a molecular recognition effect. These materials are called molecularly imprinted materials [95]. Thus far, only four papers regarding the molecular imprinting of magnetic micromotors have been published [96,97,98,99]. Regarding LDSM, there is only one work where the BiOVO_4_ semiconductor and molecular imprinting by surface polymerization were used [100].

This work developed highly homogeneous butterfly-shaped micromotors by wet chemical synthesis associated with molecular imprinting by the surface polymerization of a monomer and template (p-nitrophenol) (see Figure 8A). After the modification of the micromotors, a carbon layer was formed on the surfaces, and its optical and physicochemical characteristics were changed compared to non-imprinted analogs, namely the positive surface charge and low degree of hydrophilicity. These changes, a consequence of the success of molecular imprinting, resulted in greater activity and the consequent mobility of molecularly imprinted micromotors. Analyzing the results in Figure 8B, it is possible to conclude that molecularly imprinted micromotors presented a greater propulsion speed with the increase in the amount of the target molecule in solution. Furthermore, the objective of increasing the maximum amount of template degradation was achieved, as can be confirmed in Figure 8C. The imprinted micromotors doubled the template degradation capacity compared to non-imprinted analogs.

This way, although molecular imprinting as a technique for modifying the surface of LDSM is still new, to make them more selective and consequently more effective in their purpose, we believe this is a promising route for short-term exploration.

### 3.3. External Parameters That Affect the Propulsion of LDSM

#### 3.3.1. Type and Intensity of Light Irradiation

The choice of the type of light used to trigger LDSM has always been considered a critical factor. Some studies have highlighted different light sources, such as UV light (10–400 nm), Vis (400–760 nm), and in some cases, near-infrared (NIR) light (760–2576 nm). The choice of light source is crucial as different materials respond uniquely to different electromagnetic spectrum bands. Light intensity is also an essential and highly explored parameter, especially in the recent literature. Variation in light intensity can affect the speed and efficiency of LDSM and is, therefore, a critical aspect to be considered. Next, the three types of light will be discussed independently, highlighting the main recent activities.


**UV light**


The choice of this light source is intrinsically linked to the selection of materials sensitive to UV light, traditionally centered on metallic oxides, such as TiO_2_, which continue to be the most used in preparing LDSM with photosensitive properties in the UV range, as indicated in Table 1.

The main limitation of using UV light sources lies in biocompatibility, especially in biomedical contexts, where this light source can harm different cell lines, such as bacterial, fungi, and also tissues. Furthermore, it must be considered that using UV light to trigger movement is not ideal because it only corresponds to a narrow region of the solar spectrum (∼4%).

In general, the considerations present in the recent literature remain in line with those historically described for LDSM triggered by UV light. Research into the influence of UV light intensity and duration on the speed and effectiveness of micro/nanomotors remains the focus of this subtopic. Tuning these parameters is crucial to optimizing the performance of LDSM in various applications. More recent studies have explored the intensity of UV light as an approach to eliminate the need for chemical fuels, aiming to improve the propulsion [17,18,19,27,28,30,33,34,35,40,46]. One example is the spherical Janus TiO_2_/Au/Pt micromotor described by Chen et al. [18]. Hybrid propulsion is achieved by combining Pt and catalytic TiO_2_ surfaces without using chemical fuels. Propulsion forces are controlled by adjusting the intensity of the light, allowing for directed and controlled movement. As can be seen in Figure 9A, the micromotor is fueled by the electrochemical decomposition of H_2_O_2_, leading to a hydrogen ion gradient on the Pt side, causing self-electrophoresis and motion towards the Pt side. However, under UV illumination, photogenerated holes oxidize the peroxide fuel, creating a hydrogen ion gradient on the TiO_2_ side. This counterbalances the chemical propulsion force on the Pt side, resulting in a reversed direction of motion. The Au layer beneath the Pt surface was crucial in determining the propulsion mechanism and motor operation, avoiding the potential problems of switchable propulsion modes. This is because different chemical reactions can occur in the two hemispheres of the Janus structure, which can overlap if there is no efficient control. In a 10% H_2_O_2_ solution without light, protons gather near the Pt surface, creating a propulsion force from TiO_2_ to Pt (Figure 9B). Under UV light exposure at a power of 1 W·cm^−2^, the TiO_2_ surface generates photogenerated holes that oxidize H_2_O_2_ to produce protons. The equilibrium of these forces halts the motion (Figure 9C). By reducing the peroxide concentration to 2.5%, the photoelectrochemical process on TiO_2_ becomes dominant, causing proton accumulation on the TiO_2_ side and altering the driving force direction (Figure 9D).

The reversible “on-the-fly” optical brake is demonstrated in Figure 9E. The micromotor, composed of TiO_2_/Au/Pt, moves autonomously when the UV light is off, driven by Pt surface catalysis. Turning on the UV light initiates TiO_2_-side reactions, acting as an optical brake and reducing the micromotor speed. This brake can be turned off by switching off the light, restoring the original speed. Repeated UV exposure modulates the local proton gradient, providing real-time motion control. The micromotor’s directionality can also be changed based on the excess proton concentration on the TiO_2_ side, showcasing versatility in movement control under different conditions (Figure 9F).

In this way, the authors of this work proposed an integrated brake system generated by the external Au layer, which controls the photocatalytic decomposition reaction on the TiO_2_ side, counterbalancing the chemical propulsion force generated on the Pt side. This was associated with the regulation of irradiation intensity and the absence of chemical fuel, providing precise control of the motor movement.


**Visible light**


Vis light is a widely available energy source, which makes Vis-light-driven LDSM more accessible for practical applications. One of the advantages of this light source is its high % integration with sunlight, which can be exploited for outdoor applications. Another advantage of using Vis light is the lower toxicity in biological processes compared to UV light sources. This makes visible-light-driven LDSM more suitable for biomedical applications, where biocompatibility is crucial.

Early studies mainly focused on the selection of Vis-light-sensitive semiconductors for the efficient development of LDSM. Careful selection of these materials is essential to ensure an effective response to Vis light. In this context, the most widely used semiconductors have been CuO and BiVO_4_. Still, more recently, hybridization with two or more semiconductors has emerged as a prominent approach to achieve more efficient LDSM in response to Vis light [11,43,48,49,52,53,56]. For example, the strategy of using two metal oxides was described, one with the ability to absorb radiation in the Vis range (CuO) and the other more suitable for a specific function (photocatalytic activity), TiO_2_, to overcome the main challenge related to the efficient conversion of Vis light into motion [43].

Recently, another advantage has been highlighted, namely the improved ability of Vis light to penetrate aquatic environments. This characteristic makes visible-LDSMs suitable for applications in aquatic environments, such as micro- or nano-scale delivery systems in water [80].

As with UV light, the recent literature has focused on studying Vis light intensities to achieve improved propulsion without chemical fuels [29,31,32,39]. For example, nanocap-like motors, based on Au/TiO_2_, were proposed to obtain motors with green propulsion without the addition of chemical fuels and driven by Vis light [31]. In this work, nanocaps measuring 175 nm in diameter were described and presented a mechanism of autoelectrophoretic movement resulting from the surface plasmon resonance effect occurring between the Au and TiO_2_ layers, which predicts a rapid transfer of electrons between the two layers of the nanocaps. This phenomenon is known as the plasmonic photocatalytic effect within the realm of photocatalysis. Researchers have experimentally characterized this mechanism through techniques such as electron energy loss spectroscopy, energy-filtered transmission electron microscopy, and optical video tracking. Additionally, numerical finite-difference time-domain simulations were employed to study this mechanism theoretically. The photoactive nature of these nanocaps responded to variations in light intensity generated by an LCD lamp when turned on and off. Vis light intensity proved a relevant parameter for controlling propulsion speed in on/off tests. There was a decrease in the propulsion speed from 1.9 ± 0.5 µm·s^−1^ (light on) to 1.0 ± 0.2 µm·s^−1^ (light off), proving the effect caused by Vis light intensity in the increase in the propulsion speed of the nanomotors.


**NIR light**


Light irradiation in the NIR range represents an innovation in light-driven micro/nanomotors, especially when based on semiconductors. This approach is challenging since these LDSM, predominantly composed of metal oxides, hardly respond to wavelengths in the NIR region. The propulsion induced by NIR light is associated with thermal energy conversion phenomena, such as the photothermal effect resulting from the heat generated during the physical processes. These materials can heat up due to the absorption of NIR light and therefore generate a thermal impulse or propulsion. This irradiation is particularly valuable in biomedical applications, as the heat generated not only drives the movement of the motor but also inactivates the surrounding cells, thus providing a dual application. Deng et al. [38] described hybrid micromotors based on photocatalytic semiconductors, such as TiO_2_ and ZnO, which were capable of moving in response to NIR light. When exposed to NIR in a specific area, convection flows occur in the surrounding liquid due to the temperature gradient generated between the illuminated and non-illuminated region. This caused the micromotors to move together towards the center of convection, driven by the drag forces of these flows. Interactions between nearby micromotors, such as electrostatic attractions or diffusiophoretic repulsions, were activated or intensified due to the short distance between them. Thus, by adjusting the position of the NIR light source, it was possible to direct the propulsive movement of the hybrid micromotors. The variation in the intensity of NIR light also made it possible to control the formation of groups and the joint migration of micromotors quickly and reversibly, improving characteristics such as the response time, range, and speed of group displacement with increasing irradiation intensity. All the simple processes described in this work for optimizing movement using an ecoefficient light source have made these hybrid LDSM promising for future applications, especially in biomedicine, for drug delivery, for example.


**Effect of light intensity**


One of the main tests described in the literature for controlling the movement of LDSM is the “on/off” test of the light source. Many experiments show that the speed of movement increases proportionally with the intensity of light due to the greater speed of the chemical reaction, as we can see in Figure 10a which represents the results of one of these tests with hybrid LDSM [41].

Figure 10b shows the typical results described in this type of test. LDSM in motion driven by a light source reaches a certain speed, and when the light is off, there is a gradual speed reduction. When the light source is turned on again, an exponential increase in the LDSM propulsion speed is confirmed [18]. Another important factor to highlight is the positioning of the irradiation source, which can also be a way of controlling the direction of movement, focusing a light source on different points of the LDSM [9].

#### 3.3.2. Other External Stimuli

The combination of light with other external stimuli is nothing new when considering the use of chemical fuels. Countless studies have used specific chemical stimuli to modulate the behavior of LDSM. Specific chemicals such as H_2_O_2_ have triggered directional responses [9]. More recently, the use of these chemical fuels has been avoided and, in unavoidable cases, replaced by biofuels, such as glucose, to reduce toxicity, especially cellular toxicity [41,53].

However, as already mentioned in the introduction to this review, different external stimuli, namely magnetic, electrical, and acoustic, can be combined with light to trigger propulsion in motors. This diversity of external stimuli has recently created adaptable and versatile LDSM suitable for various applications. Precise control of these external stimuli has offered opportunities to improve the functionality and efficiency of LDSM in different contexts [28,30,37,39]. Tang et al. [30] proposed micromotors with propulsion activated by UV light and controlled by acoustic stimuli. In the study, TiO_2_–Au and Au–TiO_2_ microbowls were tested, using microbowls with the same composition but with a different concave inside, composed of Au or Ti (Figure 11A), allowing researchers to obtain different responses to light and acoustic stimuli, resulting in controlled propulsion. The acoustic flux generated by the oscillation of a microbowl drives the motor toward its outer concave surface. In this way, the micromotor moves towards the TiO_2_ side in the presence of UV light and H_2_O_2_. Light-driven propulsion occurs in the same or opposite directions as acoustically driven propulsion, depending on whether the TiO_2_ is on the outside or inside of the microbowl, respectively (Figure 11B,C). The TiO_2_–Au microbowl, propelled by acoustic forces, initially moves at a speed of 11.35 µm·s^−1^ for the first 2 s (blue track line). Upon exposure to UV light at 0.4 W cm^−2^, the green track line indicates a reduced distance, signifying an immediate decrease in speed to 5.1 µm·s^−1^ (2–4 s period) due to the competition between driving forces. Increasing the light intensity to 0.8 W·cm^−2^ leads to a reversal in the motion direction, with a speed of 21.8 µm·s^−1^ (4–6 s). Here, light propulsion prevails over acoustic propulsion, attributed to the enhanced photochemical reaction on the TiO_2_ surface. Initially, the Au–TiO_2_ microbowl moves acoustically at a speed of 5.2 µm·s^−1^ (0–2 s period) (Figure 11D,E). Upon UV light illumination, higher speeds of 14.6 and 25.3 µm·s^−1^ are achieved with light intensities of 0.4 W·cm^−2^ (2–4 s period) and 0.8 W·cm^−2^ (4–6 s period), respectively. The response to light stimulus and precise real-time modulation of speed and direction in acoustic-powered microbowls depend on the specific positions of the Au and TiO_2_ surfaces on the bowl structure. Achieving hybrid propulsion through a specific morphology, such as microbowls, opens opportunities for diverse applications, especially in biomedicine, such as drug delivery.

Magnetic control is another external stimulus used to direct LDSM by depositing magnetic materials such as Ni, Fe_3_O_4_, and NbFeB. Generally, the magnetic field is produced by two sets of coils in orthogonal directions. Applying alternating current to these coils results in a variable magnetic field that controls the direction of movement of the light-powered micro/nanomotors. Due to the extremely small size of these structures, adding magnetic materials can be challenging, sometimes leading to reduced magnetic strength. The Janus CoO–TiO_2_ micromotors are an example of hybrid micromotors with propulsion driven by Vis light, without the need for toxic fuels and movement directed by magnetic stimulus [39]. These micromotors demonstrated propulsion across the entire Vis light spectrum at an intensity 17 times lower than the average solar intensity. A significant advantage of these micromotors is their responsiveness to different wavelengths, from UV to infrared. The direction of propulsion can be changed by transitioning illumination from Vis light to UV light. Furthermore, the ability to respond to external magnetic fields induced by CoO allows for directional and effective propulsion. Following the previous study, but now to reduce costs and simplify the manufacturing process of UV-light-driven micro/nanomotors, Wang et al. prepared TiO_2_–Fe Janus micromotors based on economical metallic Fe and TiO_2_ that feature attractive photocatalytic propulsion and efficient steering control. The deposition of a thin layer of Fe to create the asymmetry necessary for the movement, instead of the traditional noble metals (Pt, Au, etc.), made producing these micromotors more economical and gave them magnetic activity. This feature proved extremely important as it guaranteed the controlled and effective propulsion of the new micromotors by adjusting the UV light intensity and using an external magnetic field [28].

Another example involves the preparation of a Au@TiO_2_-SiO_2_ nanotree to create multifunctional self-propelled micromotors responding to light, heat, and electric fields [37]. The synthesis was optimized to overcome the low yield issues reported in the literature. Gold was used to create the Janus structure essential to movement. The scale, charge, and functionalization of the nanotrees were evaluated for their impact on propulsion. This resulted in micromotors with multi-mode self-propulsion, including photochemical self-electrophoresis by UV and Vis light radiation, thermophoresis by NIR light radiation, and charge-induced electrophoresis under an electric field. Furthermore, high-throughput synthesis enabled efficient collection and recovery of Hg^2+^ from contaminated water.

Another external stimulus recently explored was the variation in the pH of the medium. Heckel and collaborators presented a simple solvothermal synthesis (Figure 12A) of well-defined and catalytically active BiVO_4_ micromotors, capable of efficient propulsion without asymmetrization associated with changing the pH of the medium [101]. These photochemically active LDSM, with a unique square-shaped structure (Figure 12B), respond to blue and UV light and can switch between different movement strategies, similar to bacterial micromotors. In other words, it was possible to alternate the movement between vertical and horizontal sliding under UV and Vis light irradiation at speeds of up to 7.0 µm·s^−1^. These behavioral changes were obtained based on external conditions, namely by changing the value of the pH of the medium, which in turn generated changes in the zeta potential on the surface of the LDSM, responsible for the variations in movement (Figure 12C). Cyclic irradiation testing was carried out and indicated that the orientation of the particles was consistent during each activation, allowing the researchers to conclude that the asymmetric restrictions on the orientation of the crystal facet in the LDSM gave rise to the type of movement. These particles showed promise for scalable production, as they do not need additional modification steps for introducing asymmetrical post-synthesis. Moreover, these uncomplicated particles dynamically adjust their swimming strategy based on external conditions. This adaptability allows them to navigate through intricate environments, a crucial characteristic for micromotors when addressing complex tasks.

## 4. Applications

Due to the special advantages of LDSM, such as autopropulsion, microscale, the easiness of light intensity control, and fast response to light irradiation, they present potential for various applications in the environment and biomedicine fields. Next, some of the most recent works with proven applications will be presented, and the main innovations described in them will be indicated for each area: the environment and biomedicine.

### 4.1. Environment

The potential of light-driven micro/nanomotors as an innovative tool to address environmental challenges has already been widely studied.

Proof-of-concept studies have demonstrated the potential of light-driven motors in environmental endeavors, including water-quality assessment and the efficient degradation of pollutants [102]. Wang and collaborators [103] in 2009 were the first to focus on the impact of pollutants on micromotor speed, laying the groundwork for motion-based detection, particularly concerning silver ions (Figure 13). This groundbreaking research has set the foundation for upcoming micromotor-based sensing protocols. In 2012, the same research group presented the initial application of functionalized tubular micromotors for dynamically removing oil from water [104]. Following this, Sanchez and Schmidt utilized rolled-up Fe/Pt tubular micromotors to degrade organic pollutants in water through the Fenton oxidation process. These early initiatives marked the first proof-of-concept applications of micromotors for water cleaning. Since then, there has been an exponential increase in the number of papers reporting on the environmental applications of micromotors.

In the realm of LDSM, their application extends to both sensing and monitoring as well as environmental remediation. A logical progression in the field, particularly since 2014, has been the development of water-fueled micromotors or the exploration of environmentally friendly, fuel-free propulsion activation means such as light or ultrasound [105]. Simultaneously, recent endeavors have aimed at enhancing the overall efficiency of micromotor-based remediation protocols by leveraging new nanomaterials like graphene.

In the following subsections, the main achievements already described in the literature will be briefly presented and the recent proven environmental applications with LDSM will be explored in greater detail.

#### 4.1.1. Monitoring and Sensing

LDSM introduce an innovative approach to real-time environmental monitoring, demonstrating significant potential in promptly detecting fluctuations, imminent risks, and ongoing remediation processes. Their value lies in continuously assessing inaccessible environments with minimal sample preparation. In this sense, two primary analytical methods that use LDSM come into play: direct optical observation and electrochemical signal variation detection [106]. The former involves monitoring variations in motion behavior or utilizing fluorescence “on-off” detection with functionalized LDSM interfering with analytes’ chemical processes. This mode presents a highly reproducible means of recognizing changes in motion behavior for detecting various analytes. On the other hand, electrochemical detection benefits from LDSM engaging in chemical reactions with analytes, offering the advantages of low reagent and sample quantities, reduced chemical waste, and higher sensitivity due to the superior control of molecular interactions at micro/nano scale levels. The widely described strategy evaluates alterations in the movement patterns of catalytic micromotors when exposed to hazardous chemicals [102,107].

More recently, integrating LDSM with fluorescent dyes or nanoparticles, such as quantum dots (QDs), provides a practical approach to creating sophisticated mobile environmental microsensors. Sánchez et al. [108] described a new “on-the-fly” chemical optical detection strategy based on the incorporation of fluorescence quantum dots on the surface of self-propelled tubular micromotors. The motion-accelerated binding of trace Hg to the QDs selectively quenches the fluorescence emission and leads to an effective discrimination between different mercury species and other co-existing ions (Figure 14A). In this case, what happens is that the outer surface that has a positively-charged layer exhibits accelerated binding to the Hg^2+^, selectively quenching fluorescence emission and enabling effective discrimination between relevant mercury species, as we can see in Figure 14B.

Another example was described by Pacheco and collaborators [109] who innovatively utilized Janus micromotors as mobile sensors for detecting toxins released by enterobacteria, recognized indicators of food contamination generally associated with exposure to contaminated water. As shown in Figure 14C, at the outset, the micromotor exhibits a robust fluorescent emission (ON microscopy image). The motion-induced binding of lipopolysaccharides (LPS) from Salmonella selectively dampens the intrinsic fluorescent emission of the micromotors (observe the absence of fluorescent emission in the LPS/OFF image). In the presence of interfering saccharides (glucose, fructose, and galactose), minimal fluorescent quenching is observed. In Figure 14D, the fluorescence images demonstrate the micromotors efficient navigation and effective fluorescent quenching in milk, mayo, egg yolk, and egg white samples previously contaminated with 1 ng·mL^−1^ of endotoxin. Despite a significant reduction in the micromotor speed from 452 μm·s^−1^ in water to 60, 58, 162, and 221 μm·s^−1^ in milk, mayo, egg yolk, and egg white, respectively, due to the high viscosity and complexity of the samples, their practical application remained unaffected. The micromotor assays were capable of detecting endotoxin concentrations as low as 0.07 ng·mL^−1^, well below the threshold considered toxic to humans (275 μg·mL^−1^).

This reliable and rapid approach represented significant potential for enhancing food contamination screening in safety and defense applications.

Despite the advances made in monitoring and sensing using LDSM, this is still an underdeveloped approach. Many other works have been recently published, however, with light-driven polymer-based motors (i.e., cellulose, chitosan, zeolite, activated carbon) with a high surface area to make the detection and/or adsorption necessary for monitoring applications more efficient [10].

#### 4.1.2. Remediation

The escalating global population and increasing industrialization pose a significant environmental threat due to the discharge of harmful substances into ecosystems. Contaminants like non-biodegradable organic dyes, toxic heavy metals, and other chemical products (e.g., personal hygiene products) necessitate effective environmental remediation methods. Commonly adopted techniques include nanoprecipitation, filtration, adsorption, and advanced oxidation processes, but these often require external agitation, making implementation challenging in remote locations. LDSM offer a promising solution, considering the characteristics of the most used semiconductors, leveraging their ability to actively move and overcome diffusion limitations. By transforming the readily accessible chemical fuel energy into mechanical motion, autonomously propelled LDSM offer a practical means to address noxious and enduring pollutants. The mechanical stirring introduced by these self-propelled synthetic motor systems facilitates effective solution mixing. The expulsion and ascent of microbubbles further contribute as supplementary mixing agents. Given that many catalytic degradation and adsorption removal processes are constrained by diffusion, the additional mechanical agitation hastens the catalytic and adsorption reactions, leading to improved degradation and removal progress.

Removing pollutants in aquatic environments is one of the most studied applications of LDSM. Its ability for light-controlled movement can be harnessed to navigate to specific areas and catalyze pollutant degradation reactions [8]. More recently, the possibility of inactivating harmful fungi present in water has been studied through the application of LDSM.

Villa and collaborators [57] presented single-component BiVO_4_ micromotors with well-defined micro/nanostructures capable of motion individually and collectively under Vis light irradiation. In this study, researchers investigated the fungicidal activity of BiVO_4_ micromotors under visible light in pure water. These micromotors approached yeast cells and adhered to their cell walls without external fuel (Figure 15A). Fluorescence microscopy revealed a significant increase in dead yeast cells after visible light irradiation, confirming the fungicidal photoactivity of BiVO_4_ micromotors. Colony-forming unit assays demonstrated a 40% decrease in yeast viability under Vis light and micromotors, showcasing their fungicidal effectiveness. The toxic action of BiVO_4_ micromotors was attributed to the photocatalytic generation of reactive oxygen species on their surface, disrupting microorganism cell membranes. Post-treatment SEM images revealed yeast cells attached to the micromotor surfaces, emphasizing their high affinity (Figure 15B). The nanoscale roughness of the motor surfaces provided favorable sites for microorganism adhesion, enhancing specific colonization. This research highlighted the potential of BiVO_4_ micromotors as a simpler and practical approach for eco-friendly water treatment, demonstrating their effectiveness against yeast cells through visible-light-induced fungicidal activity.

In addition to the study presented, much other work has focused on the possible application of LDSM in environmental remediation; however, these works have only presented a proof of concept of how beneficial the use of optimized LDSM for environmental remediation can be. No reports on the fabrication of LDSM at the industrial scale for practical wastewater treatment applications exist, to the best of our knowledge. By now, there are still a limited number of papers discussing the degradation of the LDSM during the remediation process and these articles date from before 2018 [110,111].

Despite all of this, the advances in the area of environmental remediation were most evident in the photocatalytic application, for the degradation of pollutants, mainly dyes (e.g., rhodamine B (Rh6G) and methylene blue (MB) [106]).

Next, we will highlight advances in the recent literature on LDSM applied to the photocatalytic degradation of pollutants, a major topic when LDSM are addressed.

##### Photocatalytic Degradation of Pollutants

The application of LDSM in photocatalysis is a relevant research topic, with several approaches described in the most recent scientific literature. Recalling traditional approaches to combating water pollution, such as using detergents or installing filtration membranes in contaminated areas, LDSM demonstrate a superior efficiency, requiring less energy consumption to remove biological and chemical pollutants. This efficiency is particularly crucial in places that are difficult to access by conventional methods. When introduced into polluted waters, these motors can be driven by sunlight, moving randomly and providing broad coverage in the water. Considering that many LDSM are obtained from photoactive materials, as mentioned in the previous sections, the photocatalysis area has benefited the most from these intelligent LDSM [27,33,56]. LDSM with photocatalytic properties play a dual role, taking advantage of the responsiveness and self-propulsion generated by photocatalytic reactions. These reactions, triggered by a single irradiation source, create movement and degradative action. The advantage of LDSM compared to conventional photocatalysts resides precisely in self-propulsion, which allows the more effective cleaning of contaminated water without the need for mechanical agitation or the creation of currents for sanitization [8]. Kong et al. [58] demonstrated that light-driven TiO_2_/Pt Janus micromotors show a remarkable efficacy in real-time photocatalytic degradation of 2,4,6-trinitrotoluene (2,4,6-TNT) and 2,4-dinitrotoluene (2,4-DNT) in pure water under UV irradiation compared to propulsionless photocatalyst analogs. The micromotor movement mechanism described was autoelectrophoresis, driven by asymmetric photocatalytic redox reactions that occur in TiO_2_/Pt Janus particles. Photogenerated h^+^ and e^−^ induced redox reactions on the surfaces of the micromotors, creating a local electric field that drove the micromotors and generated oxidative species capable of photodegrading 2,4-DNT and 2,4,6-TNT (Figure 16a). To validate the degradation of pollutants, it was crucial to demonstrate the rapid movement of TiO_2_/Pt Janus micromotors in the presence of high concentrations of 2,4-DNT (1000 μM) and 2,4,6-TNT (600 μM). Cyclic voltammetry was used to confirm the concentrations, revealing reduction peaks at −629 mV for 2,4-DNT and −529 mV for 2,4,6-TNT resulting from degradation pollutants previously adsorbed on the surface of TiO_2_/Pt Janus particles. Notably, mobile TiO_2_/Pt Janus micromotors demonstrated a superior degradation efficiency compared to their stationary counterparts, which is attributed to the better mixing and mass transfer in solution facilitated by their motion (Figure 16b,c).

Mou and collaborators [60] developed pure TiO_2_ micromotors with inherent asymmetry in the crystalline phases and also demonstrated the efficient ability to photodegrade Rh6G compared to analogous motionless photocatalysts.

The asymmetry tested in this work was introduced into spherical TiO_2_ particles by adjusting the calcination temperature to control the phase transition and primary grain growth, as illustrated in the Figure 17a. The average particle velocity showed a notable increase at T = 600 °C due to the emergence of phase asymmetry. It gradually increased from 10.3 to 11.7 μm·s^−1^ as T increased from 600 to 700 °C, reflecting the increasing degree of phase asymmetry as the crystalline grains expanded (Figure 17a). This study showcased the effectiveness of anatase/rutile TiO_2_ micromotors as moving photocatalysts for on-the-fly degradation of Rh6G with remarkable efficiency. The degradation process revealed that Rh6G in the medium underwent over 90% degradation in 10 min and was almost eliminated within 15 min, highlighting the high efficiency of the light-driven anatase/rutile TiO_2_ micromotors (Figure 17b,c). In contrast, Rh6G in the medium persisted even after 25 min when exposed to micromotors without movement capacity (stationary micromotors), whose motion was inhibited by the addition of 10 mM NaCl to impede their self-phoresis.

The degradation kinetics of Rh6G by both micromotors are summarized in Figure 17d,e. In the absence of micromotors and H_2_O_2_, the Rh6G degradation was negligible (k = 4.7 × 10^−3^ min^−1^), with a slow degradation rate (k = 0.082 min^−1^) observed when 2.5 wt% H_2_O_2_ was added. The presence of micromotors in the medium promoted Rh6G degradation, with light-driven micromotors exhibiting a significantly higher degradation efficiency (k = 0.26 min^−1^) compared to “stationary” micromotors (k = 0.10 min^−1^). This enhanced efficiency is attributed to the autonomous motion of the micromotors, facilitating the efficient dispersion of photogenerated reactive oxidative species (^•^OH) to degrade the target Rh6G molecules.

In terms of application, the most recent articles aim to prove the increased efficiency of the new LDSM described, in the ability to photocatalytically degrade pollutants. Similar to what was written before 2018, the recent literature has remained focused on improving the power conversion efficiency to obtain robust LDSM propulsion and expanding the usable wavelengths of light to operate LDSM with photocatalytic properties under favorable environmental conditions. The main highlight has been a focus on investigating the synergistic interactions among various parameters to broaden the scope of the desired objectives, particularly in accelerating and enhancing pollutant degradation processes. For example, Wang et al. designed the Janus TiO_2_–Fe motors combining the UV-light-activated photocatalysis and Fenton reaction remediation capability for the highly efficient oxidation removal of Rh6G in wastewater [112]. The TiO_2_–Fe micromotors swiftly propelled themselves through the photocatalytic decomposition of H_2_O_2_ over TiO_2_ under UV irradiation, simultaneously generating highly reactive oxygen species that facilitated the in situ transformation of organic pollutants into non-harmful byproducts. Notably, the combination of photocatalysis on the TiO_2_ surfaces and the photo-Fenton process on the Fe surfaces, coupled with the rapid mobility of these catalytic Janus micromotors, synergistically enhanced the degradation of organic pollutants. For proof of concept, these micromotors were applied in Rh6G degradation experiments, where the initial pH, the % H_2_O_2_, and the UV light intensity were evaluated. Figure 18A illustrates the autonomous generation of oxidative species on the UV-activated TiO_2_ surfaces, where the micromotors utilized photocatalytic H_2_O_2_ decomposition, resulting in a significant decontamination process. On the Fe surfaces, the oxidation of metallic Fe in acidic media produced ions, establishing a highly effective photo-Fenton cleaning system involving in situ generated Fe ions and hydrogen peroxide. Figure 18B shows the UV–Vis absorbance spectrum that demonstrates a significant reduction in Rh6G concentration when exposed to active TiO_2_–Fe Janus micromotors under UV light, indicating remarkably efficient degradation. Control experiments (Figure 18C) further confirmed the enhanced efficiency with the two-in-one self-propelled micromotors, incorporating both photocatalysis and photo-Fenton processes. The peak Rh6G degradation efficiency (approximately 95%) occurred at 12 min under optimal conditions. The degradation efficiency of TiO_2_–Fe micromotors surpassed that of only Fenton effects by 52-fold and exhibited a 40% improvement compared to photocatalytic degradation alone. In the absence of HCl, Fenton reactions were hindered, reducing the degradation efficiency to 70%. The comparison with static TiO_2_-Fe microparticles underscored the superiority of active micromotors, achieving a pollutant degradation efficiency 52 times higher (Figure 18C).

Another recent point in the improvement of LDSM applied to pollutant photocatalysis was the development of motors with multidirectional movement, that is, photocatalytic nanomotors with multiple modes of movement that proved to have a greater degradation capacity, as described by Li and collaborators [113]. In this work, self-propelled nanomotors with diverse modes of motion were developed, namely helical carbon nanocoils/TiO_2_ (CNC/TiO_2_) with diverse motion modes using the atomic layer deposition method. These nanomotors demonstrated effective propulsion and the real-time removal of emerging contaminants in deionized water when exposed to visible light. The focus of this research was the use of photocatalytic nanomotors with multiple movement modes (translation + rotation, agitation) for contaminant degradation, aiming to enhance the interaction with pollutants in various dimensions and thereby improve the contaminant degradation efficiency. The nanomotors’ distinctive spiral structure is primarily responsible for generating these multi-motion modes. Different sizes of CNC/TiO_2_ nanomotors operated under varied voltage conditions, resulting in distinct movement patterns. Photocatalytic tests conducted with CNC/TiO_2_ nanomotors showed a degradation capacity exceeding 50% for phenol within 30 min (15% higher than single-motion mode photocatalytic nanomotors and 37% greater than stationary photocatalytic nanomotors).

The low density of LDSM also proved the improvement in the photocatalytic degradation of dyes. Hollow mesoporous TiO_2_/Au Janus micromotors (JHP-TiO_2_-Au) exhibited elevated swimming velocities in the presence or absence of H_2_O_2_ [27]. These micromotors, propelled by autoelectrophoresis upon UV light exposure, demonstrated a threefold increase in speed compared to their solid counterparts, Janus TiO_2_/Au micromotors (JS-TiO_2_–Au). The increase in speed was attributed to the increased surface area, as a result of the hollow structure of the porous micromotors. Additionally, this study highlighted an augmented photocatalytic activity in JHP-TiO_2_–Au micromotors, leading to a faster degradation rate of MB, serving as a model pollutant. The active movement of micromotors enhanced catalytic activity, promoting better solution mixing and increased contact between TiO_2_ active sites and the MB dye. Furthermore, the hollow structure increased the photocatalytic efficiency by allowing a greater number of reflections and refractions of light in the hollow cavity of the LDSM. Degradation tests with MB solution and micromotors, along with 1.5% H_2_O_2_ under UV illumination, demonstrated a significant enhancement. The average speed of JHP micromotors in a 10–6M MB dye solution reached 16.5 ± 2.3 μm·s^−1^, 1.7 times faster than swimmers in pure water and surpassing previously reported speeds for JS-TiO_2_–Au micromotors. This acceleration was attributed to the synergistic effects of MB and the hollow porous structure. The degree of MB degradation reached 97% within 60 min, surpassing other Fenton systems utilizing other motors.

In conclusion, the findings provided insights into improving the performance of photocatalytic micromotors in the photodegradation of pollutants.

Despite the progress reported in recent years in environmental remediation through photocatalysis, there is still a significant gap in a crucial subarea: selective photocatalysis.

The design of LDSM with the ability to selectively recognize specific target molecules in complex matrices represents a considerable advantage. This idea has already been explored in other contexts, namely for motors powered by magnetic fields, but it was used only in one study with LDSM [100], as discussed in the section on functionalization (Section 3.2.3). This is an underexplored area due to the intrinsic difficulties in producing target-selective surface binding sites without compromising the photocatalytic capacity and respective self-propulsion. Despite these challenges, it is arguably a promising application for future research.

The dependence on the intensity and type of light is also a challenge to overcome when applying these devices, as it restricts their operation in environments with variable or insufficient lighting, highlighting the great need for operation using sunlight. Moreover, precise control over the direction and collective behavior in complex environments remains a crucial challenge, essential for the efficient execution of specific tasks (i.e., wastewater treatment). Addressing these issues necessitates the development of materials with enhanced efficiency in converting light energy into kinetic energy and the investigation of photosensitive materials capable of operating under a wide range of lighting conditions. Advanced strategies for navigation control and the optimization of photocatalytic systems for efficient pollutant degradation under sunlight is very important. Additionally, large-scale production and the reusability of LDSM, along with a rigorous assessment of their environmental impact and safety, are fundamental for enabling their practical application in environmental remediation.

### 4.2. Biomedicine

The application of LDSM in biomedicine has been a promising field of research, with several approaches and potential applications described in the literature. The ability to precisely manipulate micron- and nanometer-scales has made LDSM valuable for applications requiring cellular intervention. Some specific points most explored are the drug delivery, diagnosis, therapy, and cleaning of biological surfaces. LDSM have been chiefly explored as a drug delivery vehicle to deliver therapeutic substances in a targeted and controlled manner. The light-controlled navigation capability offered the possibility of targeting LDSM to specific areas of the body. Furthermore, LDSM have been proposed for diagnostic and therapeutic applications, especially in areas that are difficult to access. Their ability of light-controlled mobility has made them ideal candidates for navigating complex biological environments [10].

The application of LDSM in cleaning biological surfaces, such as removing bacterial biofilms, has also been explored. Light-controlled mobility could be harnessed to reach specific areas and perform cleaning tasks [75]. Less common but also important is the exploitation of LDSM as contrast agents in biological imaging techniques, where their mobility could provide additional information about specific areas [9].

Recent literature on LDSM applied to drug delivery and phototherapy has reported on the solution of specific challenges with innovative conditions. The main improvements reported were related to faster movement, directional control, the capacity of tissue penetration, action under sunlight and NIR, and the use of biofuels in reduced concentrations. All of these improvements, described in previous sections, present potential advantages for biomedicine applications. LDSM with self-propulsion driven by NIR light has been a main challenge. There are enormous advantages in using this type of irradiation for applications in biomedicine, considering the possible photothermal effect that can trigger a high temperature around the LDSM under NIR light irradiation and which could lead to the inactivation/death of some cells and tissues of patients [10,114,115].

The penetration/adhesion capacity of LDSM in tissue is one of the main challenges and obstacles to its application in biomedicine. To overcome this limitation, Vis-light-driven TiO_2_@N–Au nanomotors were to described for autonomous penetration by photoelectrophoresis into the vitreous body for the treatment of ocular diseases [29]. To this end, Vis light irradiation is significant due to its simplicity, precise controllability in time-space, and not being lethal to biological media. In this study, the surface doping of TiO_2_-based nanomotors with N proved sufficient to obtain the propulsion response with Vis light irradiation. Au metal was used to obtain the asymmetry necessary for propulsion. The possibility of controlling the propulsion of these nanomotors was confirmed in tests of the on/off effect of the irradiation source. The speed of the nanomotors was about 1.8 µm·s^−1^ under 100 mW·cm^−2^ Vis light irradiation. When the Vis light is turned off, the nanomotor stops moving immediately, showing only a small displacement caused by Brownian motion. This study made it possible to obtain TiO_2_@N–Au nanomotors that move in a complex and dense biological environment in response to Vis light, overcoming some of the main limitations of the most recent motors described in the literature.

Wang et al. [116] proposed a novel approach known as the biological chemotaxis-guided self-thermophoretic nanoplatform (BTCN) for autonomous mucus penetration in the targeted treatment of colorectal cancer (Figure 19A). In this nanoplatform, mesoporous silica nanoparticles were utilized as a matrix with a Pt coating on the hemispheres, creating asymmetric SiO_2_/Pt Janus structures. Subsequently, a *Staphylococcus aureus* biomimetic membrane was applied, enabling specific targeting of the intestinal inflammatory environment associated with colorectal cancer. The nanomotor exhibited asymmetrical absorption of an NIR laser, generating a self-thermophoretic effect that autonomously propelled it through the colon mucus barrier. In comparison to conventional SiO_2_/Pt nanoparticles, the bioavailability of the light-driven nanomotor increased by 2.6 times. This breakthrough provided valuable insights into the non-destructive penetration of intricate biological barriers, paving the way for effective oral targeted therapy for colorectal cancer.

Similar to previous work, autothermophoretic nanomotors, driven by NIR light, were used for phototherapy of colorectal cancer [117]. These motors showed a greater capacity to penetrate intestinal mucus and a reduction in the interception of colorectal cancer pathogenic bacteria, as seen in Figure 19B. The nanoplatform containing the LDSM was constructed using hollow mesoporous copper sulfide asymmetrically coated with titanium dioxide (CuS/TiO_2_), and the *Staphylococcus aureus* biomimetic membrane was used to disguise the nanoparticles. When subjected to NIR laser irradiation, the autothermophoresis of the nanoparticles, resulting from their asymmetric absorption of the NIR laser to create a thermal gradient, drove the motor to rapidly traverse intestinal mucus, thereby increasing the efficiency of drug delivery. The penetration efficiency of intestinal mucus increased by 2.7 times, and the interception of pathogenic bacteria decreased by 3.5 times with the movement of the motor. Light-driven nanomotors have significantly improved colorectal cancer therapy by efficiently penetrating intestinal mucus through autothermophoretic propulsion, offering a simple and universal strategy for targeted oral drug delivery.

Following a similar idea, Xing et al. [118] developed a dual-propelled nanomotor utilizing chemical mechanisms and NIR irradiation, employing CuS/Pt Janus nanoparticles encapsulated with IR820 for hypoxia relief, enhanced tumor penetration, and synergistic photodynamic and photothermal therapy (Figure 19C). The presence of copper played a crucial role in catalyzing the conversion of H_2_O_2_ into toxic ^•^OH for chemokinetic therapy. Simultaneously, deposited Pt effectively catalyzed the conversion of tumor endogenous H_2_O_2_ into oxygen, alleviating the hypoxic condition and enabling the chemical propulsion of nanomotors. The self-thermophoresis activated by NIR facilitated the uptake and penetration of 3D tumor cells.

This autonomous motion significantly improved the accumulation of nanomotors within the tumor and facilitated deeper penetration in vivo.

In vivo experiments demonstrated that the combination of Cu, chemodynamic therapy, and NIR light achieved a tumor inhibition rate exceeding 85%. Additionally, the enhanced oxygen levels promoted the generation of reactive oxygen species, further enhancing photodynamic therapy and contributing to a satisfactory antitumor effect in conjunction with photothermal treatment.

In another example, LDSM were used as a sophisticated and versatile platform capable of performing intricate tasks in living environments. In the work developed by Liu and collaborators [119], a novel approach introduced an inhibitor-conjugated NIR laser-propelled Janus nanomotor (JNM-I), for application in the modeling of amyloid-β protein (Aβ) aggregation, a phenomenon closely associated with Alzheimer disease. The Janus nanomotor (JNM) was created by sputter-coating of gold on silica template nanoparticles to form the Janus structure. The Aβ-targeting peptide inhibitor D-RK10-Cys was then attached to the gold part, resulting in JNM-I through Au−S bonds. The modification did not alter the overall structure of JNM, as seen in Figure 20A. When subjected to NIR light, JNM-I efficiently propelled itself through the self-thermophoresis effect. The active motion of JNM-I increased interactions between immobilized inhibitors and Aβ species, resulting in a heightened modulation of on-pathway Aβ aggregation. This was evidenced by distinct changes in amyloid morphology, conformation, and cytotoxicity. For instance, under NIR irradiation, 200 μg·mL^−1^ of JNM-I increased cultured SH-SY5Y cell viability from 68% to nearly 100%, whereas without NIR irradiation, it provided only 89% viability protection (Figure 20B). Simultaneously, NIR irradiation significantly enhanced the blood–brain barrier penetration of JNM-I. This innovative JNM-I successfully bridged artificial nanomotors with protein aggregation, offering new perspectives on potential applications in the prevention and treatment of Alzheimer’s disease.

Bacterial biofilms contribute to various diseases and pose challenges due to their association with resistant and chronic infections, particularly in immunocompromised individuals. Addressing this issue, nanomotors propelled by thermophoresis offer a promising solution, as they generate no waste products during propulsion. For this purpose, Maric and co-workers [120] described mesoporous silica nanoparticles coated with a thin layer of gold to create nanomotors responsive to NIR light irradiation for effective biofilm deformation (Figure 21a). The resulting mesoporous SiO_2_–Au nanomotors exhibit efficient self-propulsion under NIR exposure. These Janus nanomotors could dynamically adjust their velocity, ranging from a few µm.s^−1^ to approximately 100 µm.s^−1^, by modulating the laser power. Interestingly, altering the wavelength within the NIR spectrum did not influence the nanomotors speed change under the same laser power. SiO_2_–Au nanomotors demonstrated exceptional capability in gradually disrupting the *P. aeruginosa* biofilm after 30 s, 1 min, and 3 min of irradiation. Prolonged exposure enhanced eradication by 20% and 71% for 30 s and 3 min, respectively (Figure 21b). The nanomotors’ light-responsive and precisely controlled locomotion led to effective biofilm deformation. These findings suggest a potential platform for clinically guided therapy utilizing microscopic imaging techniques.

In conclusion, biomedicine is a very promising area for LDSM application. Many other recent articles report the possible application of LDSM in biomedicine; however, they do not present a concrete application. The transition of LDSM from the laboratory to clinical practice faces significant challenges. There is much to investigate, considering the need for ecological/biological conditions for specific applications. Since semiconductors do not always provide these characteristics, it makes sense to hybridize and combine properties with other materials to create the biological conditions necessary for applying LDSM in biomedicine. Furthermore, biocompatibility arises as a primary concern, demanding the development of LDSM from safe and biodegradable materials, as well as the optimization of their surfaces to minimize toxicity and promote cellular integration. Additionally, the complexity of the human biological environment requires advanced control systems for the precise navigation of LDSM, necessitating strategies that combine multiple stimuli and intelligent adaptation algorithms. The therapeutic efficacy of LDSM is limited by their ability to penetrate and distribute effectively through tissues, challenging the engineering of optimized designs and the use of molecular targeting strategies. The dependence on external light sources for energy and propulsion presents another obstacle, especially in deep tissues where light is limited, raising the need for innovations in energy storage and alternative propulsion methods. Lastly, rigorous regulation and clinical approval requirements necessitate detailed studies of safety and efficacy, as well as close collaboration with regulatory bodies from the early stages of development. Overcoming these challenges requires an interdisciplinary and collaborative approach, integrating advances in nanotechnology, materials science, biomedical engineering, and life sciences, to unlock the potential of LDSM in transforming biomedicine.

## 5. Conclusions

This review addressed recent advances in the field of LDSM, emphasizing the significance of photocatalytic semiconductors such as TiO_2_, ZnO, and CuO_2_ in the development of these devices. A state-of-the-art summary was provided for the period 2018–2023, highlighting the utilization of TiO_2_ for LDSM acquisition. Its widespread application was attributed to its photocatalytic characteristics, chemical stability under light irradiation, low cost, and low toxicity. To overcome its main limitation of exclusive UV light absorption, various strategies has been explored in recent years to extend absorption capabilities into the visible region. This includes combining TiO_2_ with a visible-light-absorbing photocatalyst through a p-n junction [32,39,43], surface doping [29,31], or dye sensitization [65].

The use of carbon sources, such as graphene, was discussed in this review as still being in an early stage of investigation but promising due to significant advantages such as a high surface area, good electrical conductivity, high intrinsic mobility, and excellent mechanical resistance. This material, in combination with metal oxides, has been a major focus in recent years for the development of new LDSM. Hybrid approaches combining different semiconductors (i.e., Cu_2_O@CdSe [54] or TiO_2_/MnO_2_ [16]) have demonstrated the ability to overcome individual limitations, providing a higher efficiency and directional control. The state of the art presented in this review concluded that the recent literature on LDSM has mainly focused on optimizing the efficiency and directional control of self-propulsion in different environments.

Various parameters have been scrutinized for this purpose, emphasizing the importance of two main groups of factors for optimizing LDSM propulsion: motor characteristics and environmental conditions. Structural parameters, such as morphology and surface characteristics, were identified as crucial in propulsion efficiency. Morphology, central to all micro/nanomotors, directly impacts various properties such as the propulsion efficiency, movement stability, environmental interactions, load capacity, and response to external stimuli. Spherical and asymmetrically tubular shapes, known as Janus structures, have been crucial in improving self-propulsion by allowing controlled movement direction. Janus micromotors, featuring one hemisphere responsible for propulsion and another for triggering specific activity, are typically obtained by depositing noble metals like Au and Pt to form versatile LDSM. Innovative approaches, like the deposition of metal oxides to create the “tail” of motors, exemplified by TiO_2_ on SiO_2_ motors, have proven effective in directing and enhancing the speed of light-driven micro/nanomotors.

Regarding surface characteristics of LDSM, including crystallinity, surface area, and porosity, these were identified as crucial factors influencing efficiency and properties. Innovative strategies, like the controlled introduction of different crystal facets in the same motor, were suggested to enhance controlled and directed self-propulsion [55]. Functionalization of LDSM was also highlighted as a crucial strategy to adapt motors for specific applications, improving photocatalytic efficiency and enabling response to specific stimuli. The exhibition of specific functional groups on the surface and creating complementary sites in shape and size, through techniques like molecular imprinting, has emerged as a promising approach for conferring chemical selectivity to LDSM, opening possibilities in areas such as selective drug delivery and photocatalysis. However, this is a recent and promising strategy, with only one article identified in the LDSM field [100].

Strategies for optimizing the choice of light and its repetitive intensity to trigger LDSM were also addressed, considering aspects such as biocompatibility and application specificity. Careful light selection was deemed critical, with visible and NIR light being explored more recently due to their practical and biomedical advantages. Combining light with external stimuli, such as magnetic fields [28,39] and pH variations [101], revealed innovative strategies for directing LDSM, emphasizing biocompatibility concerns when replacing chemical fuels with biofuels.

Recent LDSM have demonstrated significant potential in various environmental and biomedical applications, including environmental monitoring, signal detection, environmental remediation, selective drug delivery, and selective photocatalysis.

In the environmental realm, LDSM have emerged as innovative tools for addressing environmental challenges, excelling in water quality monitoring and efficient degradation of pollutants. In monitoring and detection, LDSM offer an innovative approach, enabling the direct optical observation and detection of electrochemical variations. The integration of LDSM with fluorescent dyes or nanoparticles, such as quantum dots, has proved adaptable for creating sophisticated mobile environmental microsensors [108]. Optical detection strategies, based on fluorescence quenching, provide a highly reproducible means of recognizing movement behavior changes to detect various analytes. In the context of environmental remediation, LDSM offer a promising solution for the effective removal of persistent contaminants such as non-biodegradable organic dyes and toxic heavy metals. The active mobility of these micro/nanomotors allows effective solution mixing, accelerating catalytic and adsorption reactions, particularly in aquatic environments. The application of LDSM in water pollutant degradation, such as inactivating harmful fungi, highlights the versatility of these devices in eco-friendly water treatment approaches [57]. Recent developments in the application of LDSM in photocatalytic degradation have emerged as a relevant research topic, with various approaches described in the latest scientific literature. Compared to traditional methods for combating water pollution, such as using detergents or installing filtration membranes in contaminated areas, LDSM demonstrates a superior efficiency, requiring less energy consumption to remove biological and chemical pollutants. This efficiency is particularly crucial in locations inaccessible by conventional methods. When introduced into polluted waters, these motors can be driven by sunlight, moving randomly and providing extensive coverage in water. However, recent articles have mainly focused on proof-of-concept studies regarding improvements obtained with new LDSM in the degradation of dyes like Rh6G and MB, compared to stationary analog photocatalysts. For example, the creation of motors with multidirectional movements and hollow structures to enhance photocatalytic efficiency was discussed. The use of helical carbon/TiO_2_ nanomotors and hollow TiO_2_/Au micromotors has demonstrated enhanced effectiveness in contaminant degradation, surpassing their stationary and solid counterparts, respectively [27]. Despite notable advancements, the practical large-scale application of LDSM for wastewater treatment has not been fully realized, and issues such as LDSM degradation during the remediation process remain areas of ongoing research. However, recent advances in environmental remediation, particularly in photocatalytic applications for pollutant degradation, indicate promising prospects for integrating LDSM into practical environmental safety and defense applications.

In the field of biomedicine, an emerging area for LDSM, recent literature has highlighted significant improvements in drug delivery and phototherapy, surpassing specific challenges with innovative conditions. Enhancements include faster movement, directional control, tissue penetration capability, and action under sunlight and NIR, as well as the use of biofuels at reduced concentrations. Self-induced propulsion under NIR light is a major challenge, offering significant advantages for biomedical applications, such as the potential photothermal effect capable of inactivating cells and tissues under NIR light irradiation.

Challenges related to LDSM penetration/adherence to tissues have been addressed with visible-light-driven TiO_2_@N-Au nanomotors, demonstrating autonomy in penetrating the vitreous humor for the treatment of ocular diseases [29]. Moreover, innovative proposals, such as using nanomotors for the effective deformation of bacterial biofilms, with controlled speed adjustment under NIR light, have proved to be a promising solution for addressing resistant and chronic infections associated with biofilms [116]. Thermally self-induced NIR-driven LDSM have also shown enhanced efficacy in penetrating the intestinal mucosa and reducing the interception of pathogenic bacteria in colorectal cancer treatment [117].

In summary, LDSM demonstrate an enormous potential in various areas of biomedicine and the environment, with recent advances indicating significant progress in overcoming challenges and expanding the possibilities for future practical applications. The need for further in-depth investigations, especially considering specific ecological/biological conditions, suggests a promising path for the hybridization/functionalization of motors and the creation of ideal biological conditions for the successful application of these LDSM in biomedicine. The main limitations pointed out continue to be the need for fuels for controlled self-propulsion, non-selectivity, and the lack of biocompatibility.


**Future Perspectives**


Future work in the field of LDSM should involve a combination of advancements in materials, manufacturing techniques, control, and integration with other technologies. As research and development progress, these semiconductor-based motors are likely to play an increasingly crucial role in various technological and biomedical applications. The enhancement of LDSM efficiency and performance will remain a continuous focus area. Optimizing the speed, directionality, and energy efficiency of these motors remain key points for more robust and effective applications. This will require choosing advanced designs with innovative materials, such as producing LDSM based on more complex geometries and incorporating hybrid materials to increase versatility and motor assembly capacity.

Continuous research into the synthesis of new semiconductor materials and the combination of different components to create more efficient and versatile motors will be a prominent approach in the near future. To address the main limitations of LDSM mentioned earlier, future perspectives include exploring functionalized LDSM for the development of new motors with specific properties, such as a better light response, increased durability, and the ability to move in complex biological environments.

Addressing one of the significant limitations of LDSM described in the literature, which is their inability to recognize a target, the production of complementary active sites in shape and available functional groups for selective recognition could be beneficial. This could enhance LDSM recognition capability and enable applications in selective photocatalysis and drug delivery. For this purpose, the implementation of molecular imprinting could be valuable and successfully achieve the greater selectivity of LDSM without the loss of other crucial surface characteristics for other applications, particularly photocatalytic ones.

Regarding the need for toxic fuels for self-propulsion, exploring asymmetric structures obtained with metals, metal mixtures, and metal oxides with magnetic characteristics (CoO, Fe_3_O_4_) may be beneficial. This could combine the self-propulsion response not only to light but also to an external magnetic field. Greater mobility in more viscous liquid environments could be achieved by exploring hollow structures to reduce densities and adjusting their surface hydrophilic/hydrophobic balance. Considering challenges beyond improving speed and self-propulsion control, the development of LDSM capable not only of interacting with the environment but also of communicating and forming complex assemblies would represent significant progress. In this case, it would be important to focus on strategies where motors can join forces to perform complex tasks, emphasizing cooperative effects. This means finding strategies for motors to cooperate with each other, forming swarms or adjusting behaviors. For this, research into different communication methods, such as chemical signaling, will be necessary.

The search for more sustainable materials and efficient production methods will continue to be an important consideration. Exploring previously unexplored metals or metal oxides (e.g., Al, Ni, Cu/Ag, CuO) to obtain asymmetric structures may be a beneficial solution considering cost reduction and eco-friendliness. Additionally, exploring more sustainable energy sources to drive LDSM could be an active research area. Research in remote control and autonomous systems will enable the more effective use of LDSM in various applications. This may include the ability to remotely control motor movements or create autonomous systems that respond to specific stimuli. For example, sensitization with specific dyes with absorption capabilities of different wavelengths in the visible region could be an alternative to control and direct interaction between different motors in the same medium. In summary, LDSM represents an emerging area that is still in an early stage and requires much future research.

## Figures and Tables

**Figure 1 molecules-29-01154-f001:**
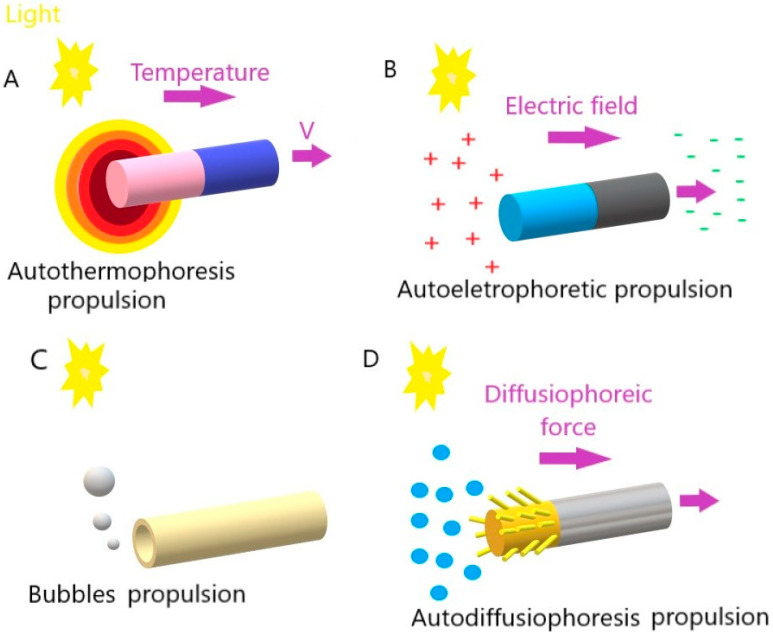
Movement mechanisms of light-driven micro/nanomotors: (**A**) autothermophoresis, (**B**) autoelectrophoresis, (**C**) bubbles, and (**D**) autodiffusiophoresis.

**Figure 2 molecules-29-01154-f002:**
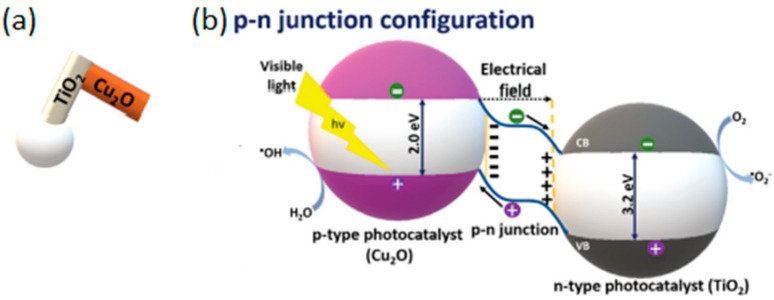
(**a**) Schematic representation of TiO_2_/Cu_2_O micromotors and (**b**) schematic illustration depicting the transfer of charges in a heterostructure with a p–n junction. Reprinted with permission from [43] (Copyright © 2018 WILEY-VCH).

**Figure 3 molecules-29-01154-f003:**
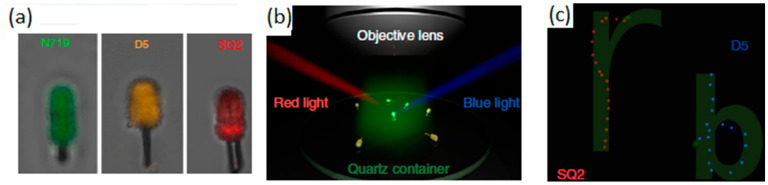
Fluorescence mapping images of the dye-sensitized nanomotors reveal the selective loading of all three dyes onto the surface of TiO_2_ nanowires (**a**); diagram of the experimental arrangement that illustrates the use of blue (475 nm) and red (660 nm) side lighting to manipulate the alignment of nanomotors (**b**); trajectory of SQ2 (red) and D5 (blue) sensitized nanomotors that naturally trace the letters ‘r’ and ‘b’ under the influence of blue and red light, respectively (**c**). Adapted from [65] (Licensed under an open-access Creative Commons CC BY 4.0 license).

**Figure 4 molecules-29-01154-f004:**
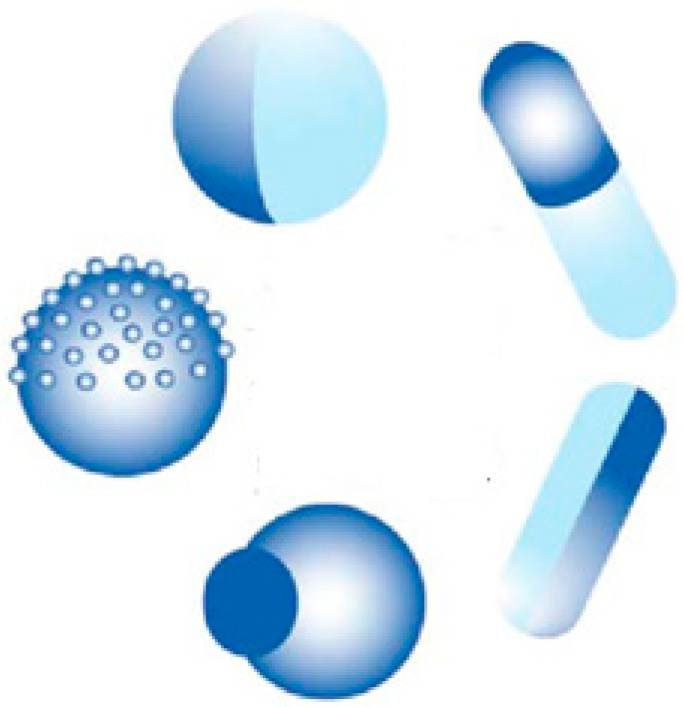
Schematic representations of Janus structures with different morphologies and shapes.

**Figure 5 molecules-29-01154-f005:**
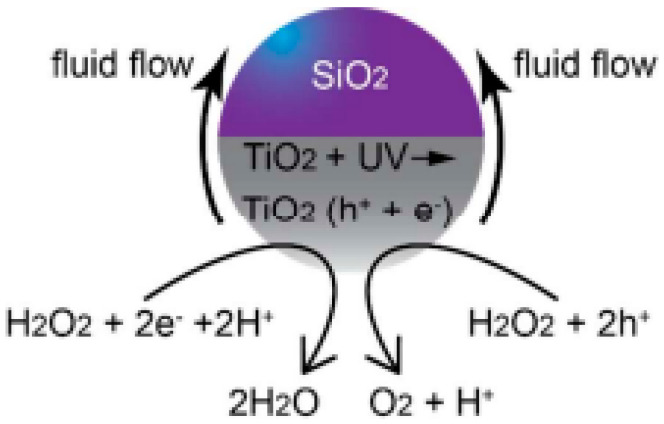
Schematic showing the possible mechanism of motion for a TiO_2_/SiO_2_ Janus sphere in H_2_O_2_. Reprinted with permission from [34] (Copyright © 2018 American Chemical Society).

**Figure 6 molecules-29-01154-f006:**
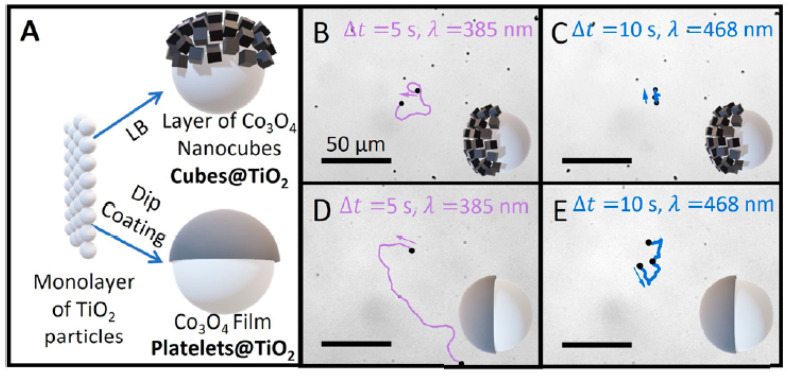
Representation of the processes involved in creating Co_3_O_4_@TiO_2_ Janus particles: (**A**) schematic representation of the fabrication methodologies; (**B**) trajectory of a cubes@TiO_2_ particle exposed to UV light with a wavelength of 385 nm (energy of 3.22 eV) for 5 s; (**C**) trajectory is extended to 10 s under blue light with a wavelength of 468 nm (energy of 2.65 eV); (**D**) trajectory displayed by a Janus particle prepared through dip coating, exposed to UV light for 5 s and blue light for 10 s, both with their respective wavelengths and energies mentioned earlier in parts (**D**,**E**). Reprinted with permission from [32] (Copyright © 2021 American Chemical Society).

**Figure 7 molecules-29-01154-f007:**
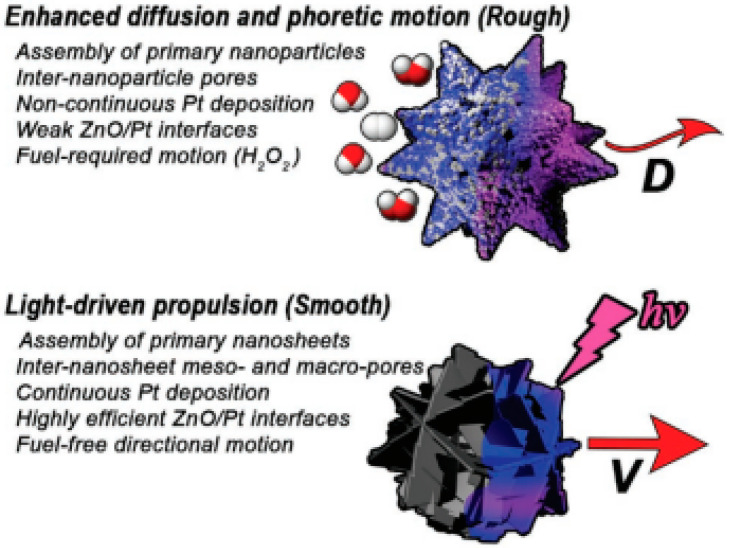
Different motion mechanisms for ZnO/Pt Janus micromotors are characterized by “rough” and “smooth” interfaces with the same specific surface area. Adapted from [50] (Licensed under an open-access Creative Commons CC BY 4.0 license).

**Figure 8 molecules-29-01154-f008:**
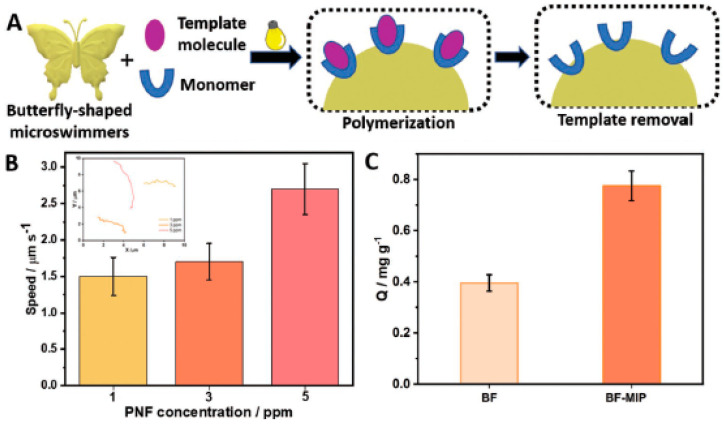
(**A**) Schematic representation of the synthesis of molecularly imprinted butterfly-shaped micromotors; (**B**) Motion speeds of the micromotors at different template concentrations; (**C**) Adsorption capacities of the template by the micromotors molecularly imprinted (BF-MIP) and analogs non-imprinted (BF). Adapted from [100] (Licensed under an open-access Creative Commons CC BY 4.0 license).

**Figure 9 molecules-29-01154-f009:**
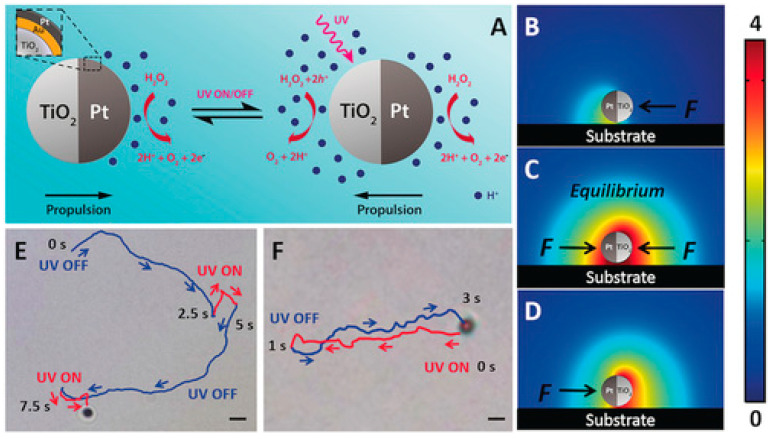
(**A**) Schematical behavior of a Janus TiO_2_/Au/Pt hybrid motors, under different irradiation conditions; (**B**–**D**) 2D simulations showing the distribution of hydrogen ions around the Janus micromotor under chemically driven motion, an “on-the-fly” brake, and light-induced reverse motion. (**E**) “On-the-fly” optical brake of a TiO_2_/Au/Pt hybrid Janus micromotor was driven chemically (blue trajectories) from 0 to 2.5 and 5.0 to 7.5 s. Turning the light on initiates the photocatalytic reaction on the TiO_2_ side, causing the micromotor to slow down at 2.5–5.0 and 7.5–9.0 s (red trajectories). (**F**) Reverse motion of a Janus micromotor propelled initially under the light-driven mode (0–1 s, red trajectory); turning the UV light off reverses the direction of the moving Janus micromotor (1–3 s, blue trajectory). Adapted from [18] (Licensed under an open-access Creative Commons CC BY 4.0 license).

**Figure 10 molecules-29-01154-f010:**
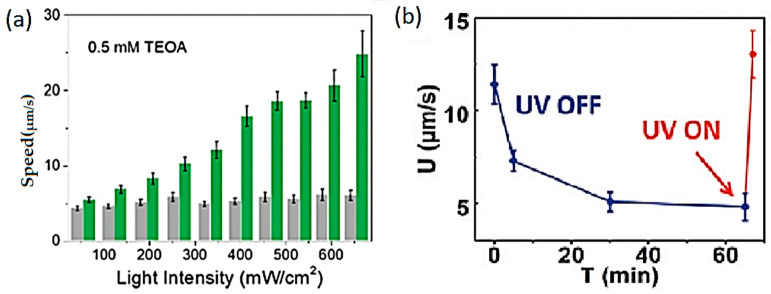
Representative examples from the literature of (**a**) the linear increase in propulsion speed with increasing light intensity. Reprinted with permission from [41]; (Copyright © 2021 Royal Society of Chemistry); (**b**) speed profile in on/off tests. Adapted from [18] (Licensed under an open-access Creative Commons CC BY 4.0 license).

**Figure 11 molecules-29-01154-f011:**
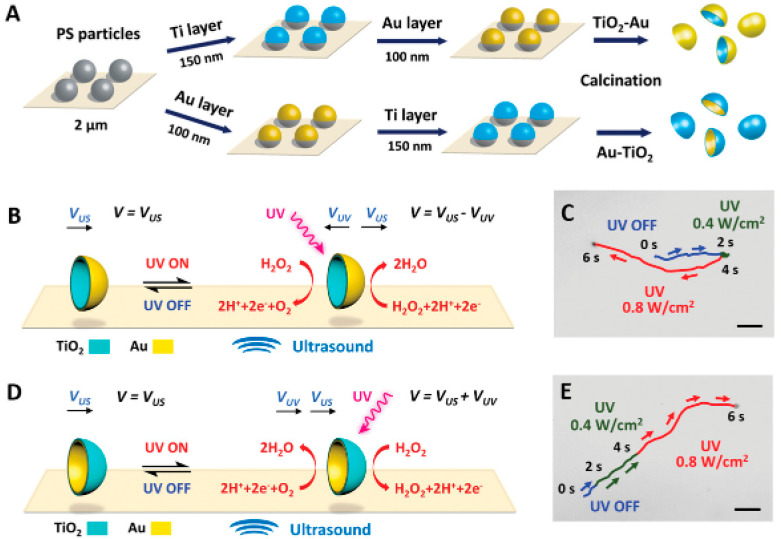
(**A**) The creation process and how the optical properties of TiO_2_–Au and Au–TiO_2_ microbowls change based on their structure when exposed to sound waves. (**B**) A TiO_2_–Au and (**D**) Au–TiO_2_ microbowl motion under the acoustic field. Continuous control of optical braking and motion direction reversal of a (**C**) TiO_2_–Au microbowl or (**E**) Au–TiO_2_ microbowl in an acoustic field (2.66 MHz, 5 V). Adapted from [30] (Licensed under an open-access Creative Commons CC BY 4.0 license).

**Figure 12 molecules-29-01154-f012:**
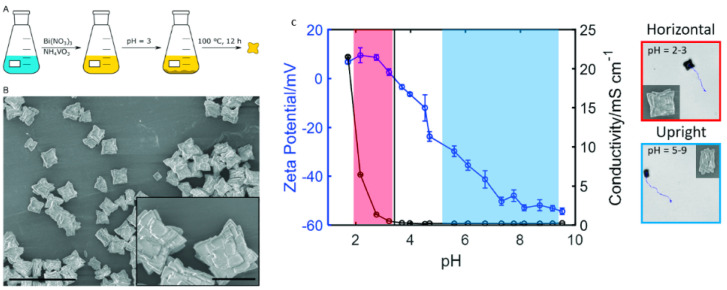
(**A**) Outline of the synthesis process: Following the dissolution of the precursors, the pH is regulated to 3 using sodium hydroxide, and a solvothermal treatment is carried out. (**B**) SEM images of the particles produced display a square-shaped, layered microstructure measuring 2 μm. (**C**) The zeta potential of the microparticles varies with the solutions’ pH. The point of zero charge is determined to be at pH  =  3.1. Below this pH, BiVO_4_ exhibits a positive surface charge, while above this pH, the surface charge becomes negative. Adapted from [101] (Licensed under an open-access Creative Commons CC BY 4.0 license).

**Figure 13 molecules-29-01154-f013:**
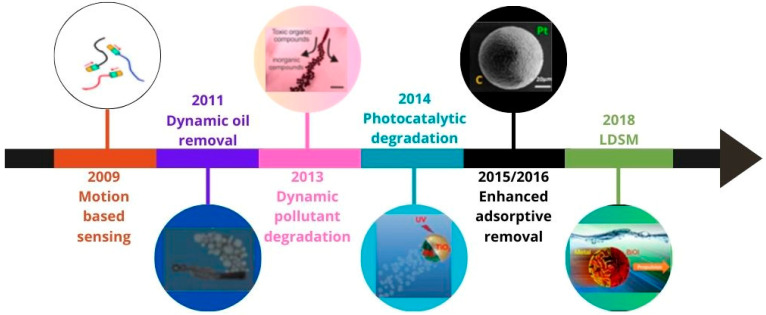
Timeline illustrating the advancements i employing autonomous micromotors for environmental purposes.

**Figure 14 molecules-29-01154-f014:**
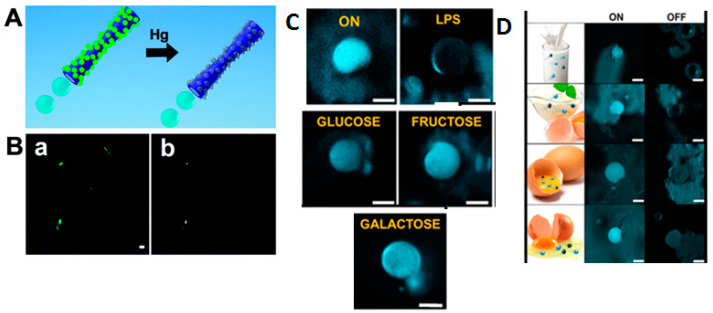
(**A**) Schematic of the micromotors and their “on-the-fly” selective detection of mercury ions based on fluorescence quenching. Fluorescence microscopy images of: (**B**) micromotors before (**a**) and after (**b**) addition of 3 mg.L^−1^ of Hg^2+^; (**C**) the micromotors before (ON) and after LPS addition (OFF) and selectivity of the protocol in the presence of interfering saccharides; (**D**) Detecting LPS in food samples (milk, mayo, egg yolk, and egg) contaminated with LPS from *Salmonella Enterica*, captured at 0 and 15 min after the microsensors had traversed the contaminated solution. Reprinted with permission from [108] (Copyright © 2018 Royal Society of Chemistry) and [109] Copyright © 2018 American Chemical Society.

**Figure 15 molecules-29-01154-f015:**
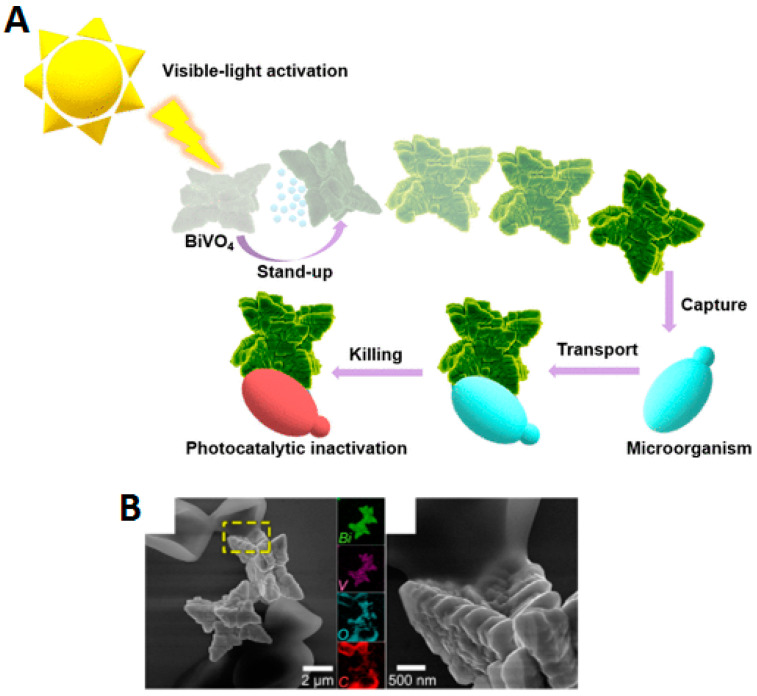
(**A**) Schematic representation of controlled cargo transportation and cells inactivation obtained by Vis-light BiVO_4_ micromotors. (**B**) SEM image that represented the interaction of the yeast cells with the surface of the BiVO_4_ micromotors after the photocatalytic treatment. Reprinted with permission from [57] (Copyright © 2019 American Chemical Society).

**Figure 16 molecules-29-01154-f016:**
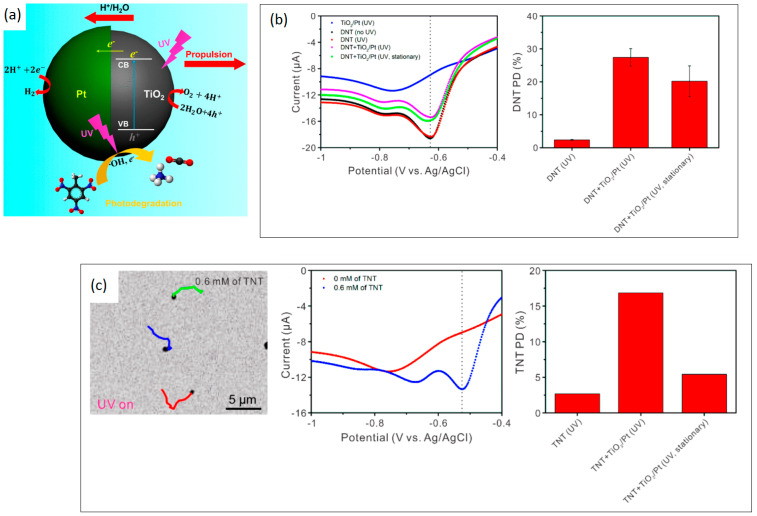
(**a**) Representative illustration the propulsion mechanism of UV-light-powered TiO_2_/Pt Janus micromotors and photodegradation of nitroaromatic molecules. Cyclic voltammograms and evaluation of the degradation efficiency, under UV irradiation of (**b**) 2,4-DNT (100 μM) without TiO_2_/Pt Janus micromotors; 2,4-DNT + TiO_2_/Pt Janus micromotors (0.12 mg·mL^−1^), and 2,4-DNT + TiO_2_/Pt particles (without movement), (**c**) the same results for 2,4,6-TNT. Reprinted with permission from [58] (Copyright © 2018 American Chemical Society).

**Figure 17 molecules-29-01154-f017:**
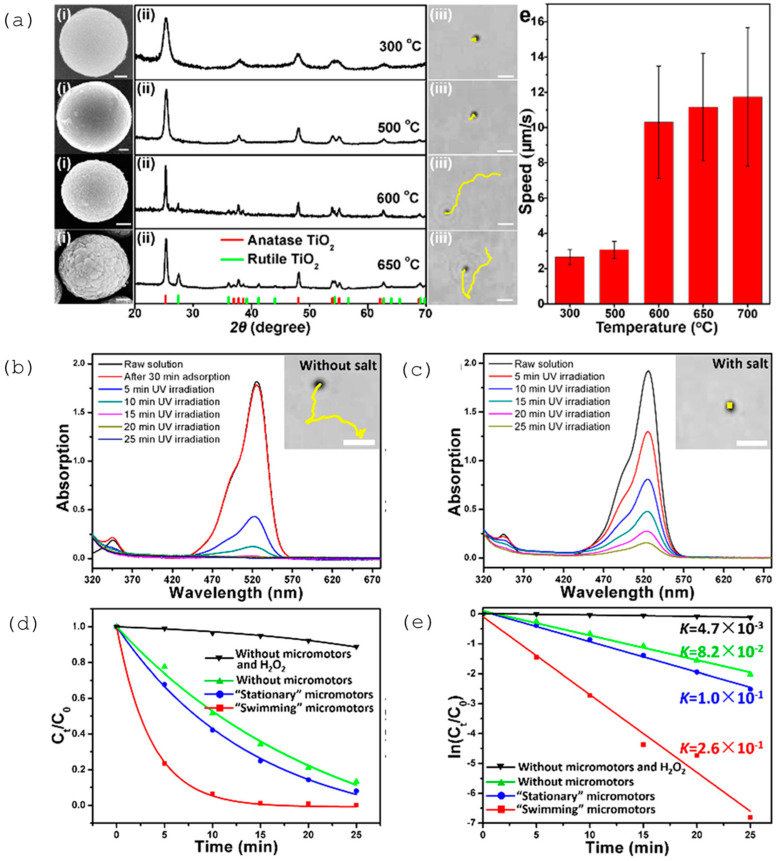
(**a**) SEM images (**i**), XRD patterns (**ii**), and motion trajectories over 2 s (**iii**) were captured under UV irradiation for TiO_2_ micromotors prepared at different temperatures. The speed of the micromotors, obtained at different temperatures, indicates an increase with T under UV irradiation (I = 80 mW·cm^−2^). Moving on to the photocatalytic degradation of Rh6G, absorbance spectra of Rh6G in the medium (25 μM) before and after UV exposure (1.8 W·cm^−2^) for different times were recorded with both “swimming” (**b**) and “stationary” (**c**) micromotors obtained at T = 700 °C. (**d**) Variations over time in the equilibrium concentration of Rh6G in a solution due to the photocatalytic action of “swimming” (represented by red squares) and “stationary” (represented by blue dots) micromotors. (**e**) Degradation rate constants (k) determined by fitting experimental data to a first-order kinetic model. To establish controls, Rh6G degradation was also observed in medium without micromotors and H_2_O_2_ (indicated by black triangles) or with H_2_O_2_ alone (shown by green triangles). Reprinted with permission from [60] (Copyright © 2019 American Chemical Society).

**Figure 18 molecules-29-01154-f018:**
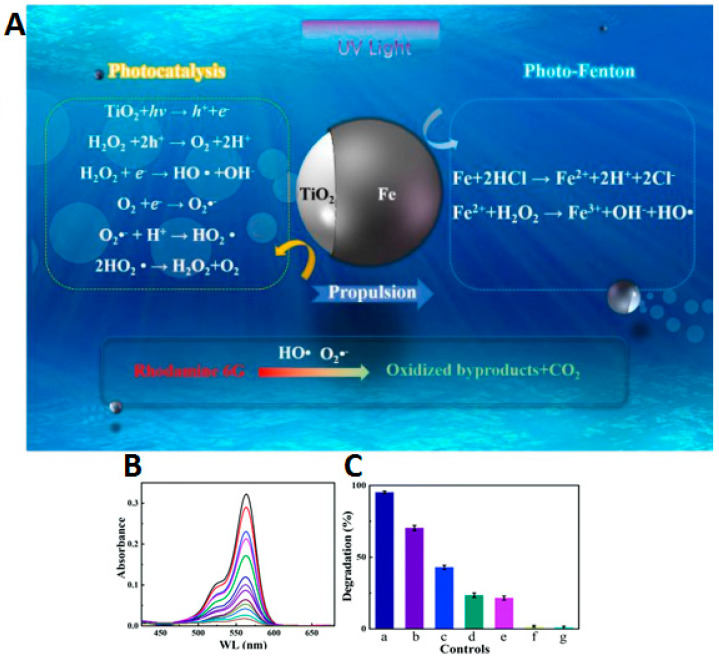
(**A**) Representative scheme of mechanism involved in the TiO_2_–Fe micromotors performing the degradation of contaminated water (using Rh6G as a model contaminant) through a combination of photocatalytic degradation and the Fenton process, resulting in the production of inorganic products. (**B**) Absorbance spectra of Rh6G are monitored over time in the presence of micromotors under UV irradiation at pH 2.5. The experimental setup includes Rh6G (initial concentration C_0_ = 5 mg L^−1^), 5% H_2_O_2_, and 1% SDS as a surfactant in a total volume of 600 μL. (**C**) The efficiency of Rh6G photodegradation with micromotors under various conditions: (a) Rh6G + TiO_2_–Fe micromotors + H_2_O_2_ + UV + HCl, (b) Rh6G + TiO_2_–Fe micromotors + H_2_O_2_ + UV without HCl, (c) Rh6G + TiO_2_ + Fe + H_2_O_2_ + UV + HCl, (d) Rh6G + TiO_2_–Fe micromotors + UV + HCl without H_2_O_2_, (e) Rh6G + TiO_2_ + H2O2 + UV + HCl, (f) Rh6G + TiO_2_–Fe micromotors + H_2_O_2_ + HCl without UV, (g) blank control (Rh6G + H_2_O_2_ + UV + HCl). Reprinted with permission from [112] (Copyright © 2019 RSC Pub).

**Figure 19 molecules-29-01154-f019:**
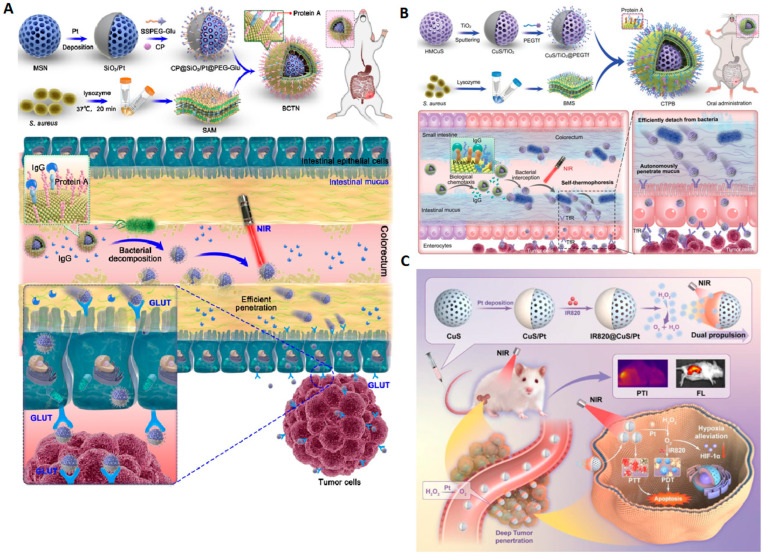
Schematic representation of (**A**) motor synthesis and respective action in therapy of BCTN; (**B**) preparation process and therapeutic mechanism of self-thermophoretic light-driven nanomotors involving cascade processes: (1) biomimetic chemotactic colorectal cancer segment colonization; (2) self-thermophoresis-driven detachment of intestinal pathogenic bacteria; and (3) autonomous penetration of intestinal mucus; (**C**) the fabrication process for dual-powered IR820@CuS/Pt nanomotors and the underlying mechanism driving deep tumor penetration, the alleviation of tumor hypoxia, and the synergistic effects of photodynamic therapy and photothermal therapy. Adapted from [116] (Licensed under an open-access Creative Commons CC BY 4.0 license); Reprinted with permission from [117,118] (Copyright © 2019 RSC Pub).

**Figure 20 molecules-29-01154-f020:**
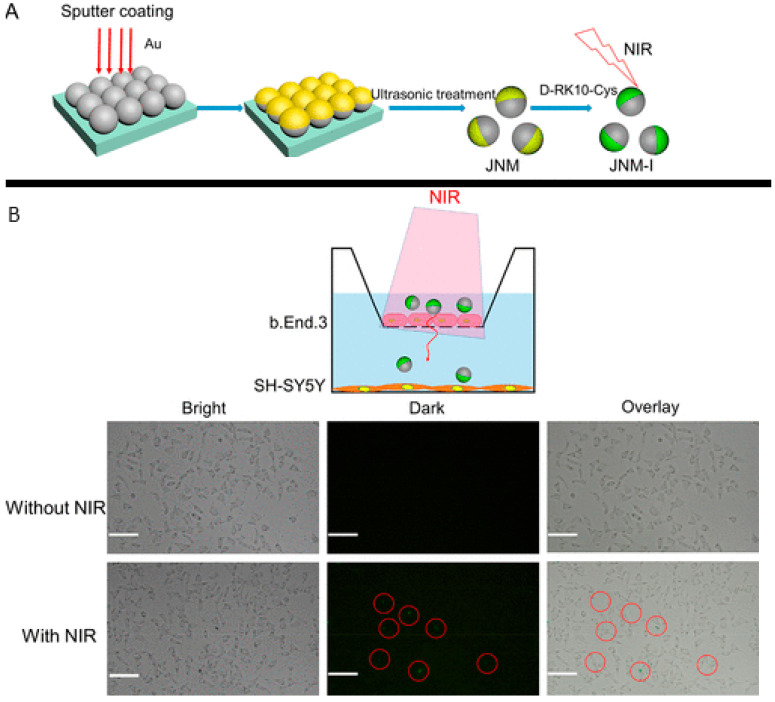
(**A**) Schematic illustration of the preparation of JNM-I. (**B**) In vitro blood–brain barrier penetration of JNM-I. The results were observed by a fluorescence microscope. Scale bar: 200 μm. Reprinted with permission from [119] (Copyright © 2020 American Chemical Society).

**Figure 21 molecules-29-01154-f021:**
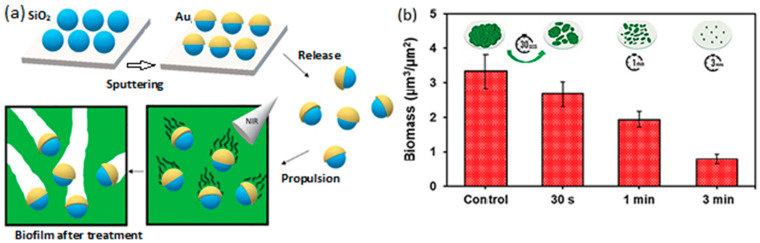
(**a**) Scheme on the production of SiO_2_/Au nanomotors, their movement when exposed to NIR light, and their role in eliminating biofilm. (**b**) Comparison of biomass levels before and after being treated with SiO_2_–Au nanomotors while exposed to NIR light for 30 s, 1 min, and 3 min. Adapted from [120] (Licensed under an open-access Creative Commons CC BY 4.0 license).

**Table 1 molecules-29-01154-t001:** Compilation of recent literature data on the production and performance of LDSM.

Chemical Composition	Propulsion Mechanism	Morphology	Scale (nm)	Light Range and Maximum Intensity (mW.cm^−2^)	Velocity [μm.s^−1^]	Ref.
TiO_2_/Au motors	Self-electrophoretic	Janus hollow sphere	~800	UV (16)	48.5	[27]
TiO_2_-Fe motors	Self-electrophoresis	Janus sphere	~2000	UV (21)	7.8	[28]
TiO_2_@N-Au motors	Self-electrophoresis n	Wire	235 ± 35	Vis (100)	2.5 ± 0.4	[29]
TiO_2-_Au motors	Self-electrophoresis and acoustic effect	Capsule	~2150	UV (80)	39.5	[30]
2D TiO_2_ motors	Bubble-propelled	Tubes	150	UV (50)	52.0	[17]
Au/TiO_2_ motors	Self-electrophoresis	Capsule	175	Vis (100)	1.9 ± 0.5	[31]
Hybrid TiO_2_/Au/Pt motor	Self-diffusiophoresis	Janus sphere	~1000	UV (10)	13.0	[18]
Cubes Co_3_O_4_ @TiO_2_ and Platelets Co_3_O_4_@TiO_2_	Brownian diffusion and self-electrophoresis	Janus sphere	900	UV (20) Vis (16)	Cubes—5.0 and 10.0 Platelets—13.9 and 19.3 for Vis and UV light	[32]
Au/TiO_2_ motors	Self-diffusiophoresis	Pillars	1500	UV (320)	4.0	[19]
Hydrogel-based motors	Bubble-propelled	Spheres	200–370	UV (300)	100.0 to 135.0	[33]
Hybrid TiO_2_@SiO_2_ motors	Self-diffusiophoresis	Janus spheres elongated	3200	UV (20)	5.0 ± 2.0	[34]
Hybrid light/acoustic-powered motors	Self-diffusiophoresis	Janus spheres elongated	1500	UV (50)	27.0 ± 9.0	[35]
Cu@TiO_2_ and Au@TiO_2_ motors	Self-electrophoresis and diffusiophoresis	Janus spheres	700	UV (315)	Au = 23.0 Cu = 38.0	[36]
Au@TiO_2_-SiO_2_ motors	Self-electrophoresis	Janus spheres elongated	250	NIR (10)	0.7 ± 0.1	[37]
*TiO_2_/Pt; SiO_2_/Pt* *ZnO/Pt motors*	Self-electrophoresis	Janus spheres	~3000	NIR (10)	21.0 to 30.0	[38]
Magnetic CoO–TiO_2_ motors	Self-electrophoresis	Janus sphere	~4000	UV (100) Vis (6)	Vis = 11.5 ± 0.7 UV = 6.2 ± 0.2	[39]
TiO_2_/MnO_2_ motors	Bubble-propelled	Janus sphere	240	UV (20)	48.1	[16]
Metal/TiO_2_ motors ((Pt, Au, Ag, Fe, Cu)	Self-electrophoresis	Janus sphere	2200	UV (40)	Pt = 8.9; Au = 2.1 Ag = 3.6; Fe = 4.3 Cu = 5.1	[40]
TiO_2_–Pt motors	Self-electrophoresis	Janus sphere	1200	UV (123)	35.0	[41]
TiO_2_ motors	Self-electrophoresis	Hollow spheres	~1000	UV–LED (590)	6.5 to 9.5	[42]
Au/TiO_2_/Cu_2_O hybrid motors	Self-electrophoresis and diffusiophoresis	Spheres elongated	~5000	UV–Vis (600)	Value not calculated	[43]
dielectric SiO_2_/TiO_2_ motors	Self-electrophoresis	Spheres	~1000	UV (35)	8.6	[15]
MOFs funcionalizado	Self-electrophoresis	Spheres elongated	2600	UV–Vis (14)	40.0	[44]
Au-ZnO motors	Self-electrophoresis	Rods	90	UV (250)	24.2	[45]
ZnO/Pt motors	Self-electrophoresis	Janus spheres	~2000	UV (135)	32.0	[46]
Sb_2_Se_3_/ZnO motors	Self-electrophoresis	Wire	100	UV (200)	3.9	[47]
ZnO/ZnO_2_/Pt motor	Self- diffusiophoresis and bubble-propelled	Janus spheres	5000	Vis (20)	352.0	[48]
Au/ Fe_3_O_4_ motors	Self-electrophoresis and diffusiophoresis	Rods	200	Vis (33)	37.2	[49]
ZnO/Pt motors	Self-electrophoresis diffusiophoresis	Sheets	~1000	UV (400)	2.5	[50]
Graphene Aerogel motors	Self-electrophoresis	Janus spheres	100–1000	NIR (9)	17.6	[51]
GQDs/PtNPs motors	Self-diffusiophoretic	Tubs	nano	Vis (N/A)	440.0	[52]
Cu_2_O@GO motors	Self-electrophoresis/ diffusiophoretic and thermophoresis	Spheres	~2000	Vis (32) NIR (2.5)	16.6	[13]
Cu_2+1_O motors	Self-diffusiophoretic	Spheres	~1000	Vis (84)	107.0	[53]
Cu_2_O@CdSe motors	Self-diffusiophoretic	Octahedral structure	~1000	UV–Vis (180)	55.1	[54]
Cu_2_O motors	Self-diffusiophoretic	Octahedral structure	864	UV–Vis (180)	10.8	[55]
GO/BiVO_4_ motors	Self-diffusiophoretic	Spheres	micro	Vis (180)	25	[56]
Single-component BiVO_4_ motors	Self-diffusiophoretic	Janus star	4000–8000	UV–Vis (5)	5 ± 1	[57]
TiO_2_/Pt motors	Self-electrophoresis	Janus sphere	~1 μm	UV (40)	9.7 ± 1.98	[58]
Au-Cu motors	Bubble-propelled	Tubes	~1 μm	NIR (6.4)	2.2 ± 0.3	[59]
TiO_2_ crystalline motors	Self-electrophoresis/ diffusiophoretic	Janus sphere	500 nm	UV (80)	11	[60]
WO_3_ motors	Self-diffusiophoretic	Spheres	1-2 μm	UV (160)	2.0 ± 0.1	[61]

N/A—Not applicable.

## Data Availability

Not applicable.

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
