# Peer review of "Recent Advances in Light-Driven Semiconductor-Based Micro/Nanomotors: Optimization Strategies and Emerging Applications"

_molecules, 2024, doi:10.3390/molecules29051154_

Round 1

Reviewer 1 Report

Comments and Suggestions for Authors

It is a very timely and high-quality review paper. Semiconductor-based colloidal motors can be modulated well by use of light illumination, which may be served as good physical models of non-equillibrium complex systems. This manuscript is organized and written well. In this case, I am happy to recommend it to be accepted for publication as it is. 

Author Response

The reviewer did not suggest any changes.

Reviewer 2 Report

Comments and Suggestions for Authors

This review mainly focus on recent advances in semiconductor-based micro/nanomotors that propelled by light. The structure of the whole paper are well organized. I believe this review will greatly help the readers to learn the advances of this research area. The following questions need to be solved.   

1. Point 3.1.4 Crystallinity, the authors concluded that  materials with a robust crystalline structure can better resist harsh environmental ocnditions..., please provide related evidences.

2. Some semiconductor materials are missing, such as WO3 (such as: Metal oxide single-component light-powered micromotors for photocatalytic degradation of nitroaromatic pollutants. npj Clean Water 6, 21 (2023)).

3. In hybrid light-driven part, integrating of other 2D materials, such as MXene ( ACS Applied Materials & Interfaces 2024 16 (1), 1293-1307.) also need to be discussed.

4. The description of Janus structures of micro/nanomotors are a little repeated, in the part of TiO2-based micro/nanomotors and asymmetric structure(Janus structure).

5. In the title, the Micromotors need to be replaced by Micro/nanomotors

Reviewer 3 Report

Comments and Suggestions for Authors

This review provides a novel perspective and valuable insights into the field of light-driven semiconductor-based micro/nanomotors. The authors not only summarize the latest developments in this field but also extensively discuss the key factors influencing the performance of these micro/nanomotors, as well as their potential in environmental and biomedical applications. However, given the rapid development and emerging new research in this field, I suggest further enhancing the timeliness and comprehensiveness of the literature review. Detailed reviewer comments are as follows:

1.      I suggest supplementing the driving mechanism section with, for example, photoinduced deformation and light-induced surface tension variation. Enriching the driving mechanisms can provide readers with a new perspective to understand another possible way of light-driven nanomotors.

2.      Although the paper has already discussed the potential applications of light-driven semiconductor-based micro/nanomotors in the fields of environment and biomedicine, it is recommended to further explore potential challenges and solutions, as well as technical issues that may arise in the practical application of these micro/nanomotors. For example, in biomedical applications, light-driven micro/nanomotors may face challenges such as biological contamination and high-salt environments.

3.      There are errors in the symbols used in lines 1029 (“as represented in Figure 6A)”) and 1093 (“primary nanoparticles and nanosheets (smooth and rough),”).

4.      In the section of “Light-driven TiO2-based micro/nanomotors”, In addition to the strategies mentioned in the article, reducing titanium dioxide to black titanium dioxide can also decrease the energy bandgap to enhance titanium dioxide absorption in the visible light region.

Round 2

Reviewer 3 Report

Comments and Suggestions for Authors

No further revisions are needed.